# A triple increase in global river basins with water scarcity due to future pollution

Mengru Wang ⬡ [1] ✉, Benjamin Leon Bodirsky ⬡ [2], Rhodé Rijneveld ⬡ [1], Felicitas Beier ⬡ [2,3], Mirjam P. Bak ⬡ [1], Masooma Batool ⬡ [4], Bram Droppers ⬡ [5], Alexander Popp ⬡ [2], Michelle T. H. van Vliet ⬡ [5] & Maryna Strokal ⬡ [1]

Water security is at stake today. While climate changes influence water availability, urbanization and agricultural activities have led to increasing water demand as well as pollution, limiting safe water use. We conducted a global assessment of future clean-water scarcity for 2050s by adding the water pollution aspect to the classical water quantity-induced scarcity assessments. This was done for >10,000 sub-basins focusing on nitrogen pollution in rivers by integrating land-system, hydrological and water quality models. We found that water pollution aggravates water scarcity in >2000 sub-basins worldwide. The number of sub-basins with water scarcity triples due to future nitrogen pollution worldwide. In 2010, 984 sub-basins are classified as water scarce when considering only quantity-induced scarcity, while 2517 sub-basins are affected by quantity & quality-induced scarcity. This number even increases to 3061 sub-basins in the worst case scenario in 2050. This aggravation means an extra 40 million km² of basin area and 3 billion more people that may potentially face water scarcity in 2050. Our results stress the urgent need to address water quality in future water management policies for the Sustainable Development Goals.

Water is an essential resource for our life and nature. Yet only 0.02% of the water on Earth is available to people, plants, and animals. Water availability is mostly assessed through flows from rivers to seas, with an estimated global annual discharge of 45,500 km³/year, largely depending on the spatial and temporal distribution of precipitation and evaporation[1,2]. Current global annual withdrawals are lower than global annual discharge. However, their spatial and temporal variations cause a mismatch, leading to water scarcity among regions[1–5].

Water scarcity generally refers to the condition wherein the water availability cannot meet the demand of nature and society[6]. Water scarcity is expected to be exaggerated in the future, being largely affected by both climate and socio-economic changes. Climate change alters spatial and temporal patterns of the hydrological cycle, leading to changes in water availability, such as river discharge[7,8]. For example, Hagemann, et al.[9] state that in 2100, water availability is expected to increase in many river basins but also severely decrease in other river basins due to climate change. Furthermore, socio-economic changes, such as land-use change, irrigation, and dam constructions, directly affect the hydrological cycle by altering the timing and magnitude of water discharge[10,11]. Fekete et al.[12] state that direct anthropogenic alterations in basins of major economic areas, such as India and China, exceed the effect of climate change considerably, leading to a larger decline in runoff in the future. In addition, socio-economic changes also affect water demand. Population and economic growth have been the main drivers for growing food demand, increased living standards, changing food and energy consumption patterns, and expansions of

[1]Earth Systems and Global Change Group, Wageningen University & Research, Droevendaalsesteeg 3, 6708 PB Wageningen, The Netherlands. [2]Potsdam Institute for Climate Impact Research (PIK), Leibniz Association, Telegrafenberg A56, 14412 Potsdam, Germany. [3]Humboldt University, Thaer-Institute of Agricultural and Horticultural Sciences, Invalidenstr. 42, 10099 Berlin, Germany. [4]UFZ-Helmholtz Centre for Environmental Research, Department of Computational Hydrosystems, Leipzig, Germany. [5]Department of Physical Geography, Utrecht University, PO Box 80.115, 3508 TC Utrecht, the Netherlands. ✉ e-mail: mengru.wang@wur.nl

irrigated agriculture. These changes have led to ever-increasing global water demand, causing water scarcity[3,13,14].

Until a few years ago, global water scarcity assessments focused mainly on changes in the quantity perspective of water availability[4,6,8,11,15,16]. However, decreasing water quality caused by increasing and newly emerging pollutants also became an important reason for limiting water to be safely used by nature and humans, aggravating the water scarcity problems. For example, agricultural intensification and urbanization have added excessive pollutants such as nutrients, pathogens, plastics, and other chemicals to the water bodies[17–21]. Among the pollutants, excessive nitrogen (N) inputs to aquatic ecosystems can have negative consequences such as harmful algae blooms, hypoxia, and fish kills and complicate the use as drinking water[22–25]. This leads to an aggravation of water scarcity because pollution limits water to be safely used.

Two studies have quantified regional water scarcity by innovatively integrating assessments of both water quality and quantity[26,27]. Van Vliet et al.[28,29] have been the first to assess water scarcity on a global scale using a sector-specific approach focusing on multiple pollutants including nitrogen for the historical period of 2000–2010. In their studies, water scarcity is assessed as the ratio of sectoral water withdrawals of acceptable water quality to the overall water availability taking into consideration of environmental flow requirements (EFRs – water flows to sustain freshwater and estuarine ecosystems). To our knowledge, there is a lack of global assessment that quantifies future water scarcity based on nitrogen pollution in rivers under climate and socio-economic scenarios. Such an assessment is essential because a better understanding of future global hotspots of water scarcity under socio-economic and climate changes will contribute to formulating effective water management policies[6,28,30,31]. Subsequently, this facilitates the supply of clean water for all, one of the United Nations Sustainable Development Goals (SDGs) for 2030[32].

This study aims to assess future global clean-water scarcity in 2050 under climate and socio-economic changes. 'Clean-water scarcity' is assessed with two indicators: a water quantity-based and a water quality-based indicator (see Methods). We define 'clean-water scarcity' as the availability of surface water with acceptable quality. Our assessment is done for >10,000 sub-basins worldwide based on their river discharges (water quantity) and nitrogen pollution levels (water quality). To this end, we combine the MARINA-Nutrients (Model to

Assess River Inputs of pollutaNts to seAs), MAgPIE (Model of Agricultural Production and its Impact on the Environment), and VIC (Variable Infiltration Capacity) models into an integrated modeling framework (Fig. 1, Methods). Results of this modeling framework are used to calculate the indicators for 'clean-water scarcity' for 2010 and 2050 under three scenarios assuming different storylines of climate changes and socio-economic activities that affect water scarcity (see Methods). Next, we identify the future global hotspots of severe clean-water scarcity and whether this is mainly driven by water quantity or water quality issues (i.e., nitrogen pollution). Last, we discuss the interactions between water scarcity, food production, and society (i.e. population and sewage) in the hotspots, taking the perspective of achieving the SDGs in these hotspots. Our results contribute to a better understanding of future water scarcity caused by changes in both water availability and water pollution. The hotspot analysis also facilitates proactive water management strategies for sub-basins where water scarcity will be potentially high in the future.

## Results
### Water scarcity hotspots of clean water
We find that current and future water scarcity becomes a substantially more severe issue globally when implementing our clean-water scarcity assessment. The number of sub-basins facing severe scarcity doubles in 2010 and may even triple in 2050 in our clean-water scarcity assessment, compared to the classical water scarcity assessment that only considers water availability from the quantity perspective (Fig. 2). Due to their high nitrogen pollution levels, many sub-basins in South China, Central Europe, North America, and Africa become water scarcity hotspots. This also implies more than a doubling of the global area and population affected by severe water scarcity in both 2010 and 2050, meaning that up to 40 million km² of extra global drainage area, including highly biodiverse aquatic ecosystems and an additional 3 billion people are facing water scarcity challenges due to nitrogen pollution (Table 1).

In 2010, one-fourth (2517 out of 10,226) of the global sub-basins face severe scarcity of clean water, according to our assessment (hotspots, Table 2). These hotspot sub-basins are mainly distributed in southern parts of North America, Europe, parts of Northern Africa, the Middle East, Central Asia, India, China, and Southeast Asia (Fig. 2). These sub-basins cover 32% of the global land area.

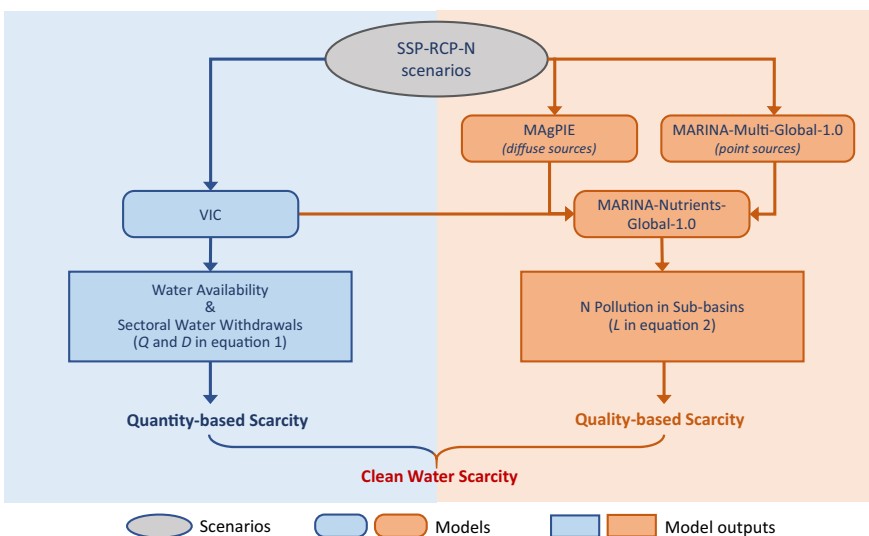

**Fig. 1 | A modeling framework for the 'clean-water scarcity' assessment in our study.** SSP-RCP-N is the scenario along the Nitrogen futures in the Shared Socio-economic Pathways[79] developed based on the Shared-economic pathways (SSPs) and Representative Concentration Pathways (RCPs). VIC is the Variable Infiltration Capacity model. MAgPIE is the Model of Agricultural Production and its Impact on the Environment. MARINA-Nutrients is the Model to Assess River Inputs of pollutaNts to seAs.

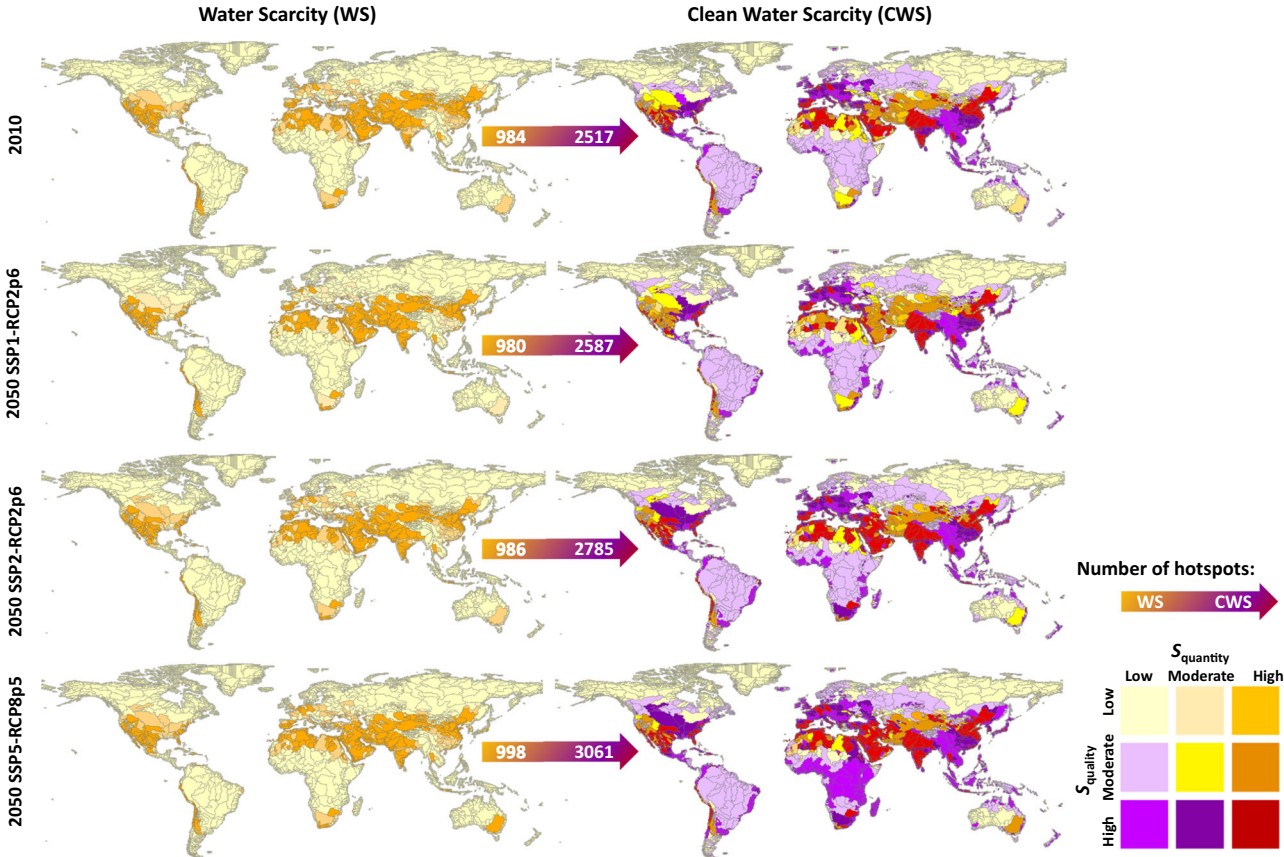

**Fig. 2 | 'Classical water scarcity' (WS) and 'Clean-water scarcity' (CWS) assessments at the sub-basin scale in 2010 and 2050.** Classical water scarcity assessment is only based on water quantity ($S_{quantity}$), while clean-water scarcity assessment is based on both water quantity ($S_{quantity}$) and quality ($S_{quality}$). Equations on how $S_{quantity}$ and $S_{quality}$ are calculated are available in the Method section. The numbers in the arrows show the number of hotspots for classical water scarcity and clean-water scarcity assessment. Hotspots are sub-basins where either $S_{quantity}$ or $S_{quality}$ or both are considered high in Table 2. For 2050, water scarcity is calculated for three scenarios: SSP1-RCP2p6, SSP2-RCP2p6, SSP5-RCP8p5. Details of the scenarios based on the Shared-economic pathways (SSPs) and Representative Concentration Pathways (RCPs) are available in Supplementary Tables S9–S11 in the Supporting Information.

About 80% of the total population lives there, contributing to 84% of global total nitrogen (N) losses to rivers from human waste. Agricultural activities are usually intensive in these regions. They cover 44% of the global agricultural land, receive 84% and 53% of the global N applications from fertilizer and manure, and produce 69% of N in global harvested crops.

The majority (2218; 88%) of the hotspot sub-basins experience clean-water scarcity dominated by nitrogen pollution. Quality-based water scarcity hotspots occurred in southern parts of North America, Europe, the Middle East, Southeast Asia, China, India, and parts of Northern Africa (Fig. 2 and Supplementary Fig. S3). These sub-basins cover 24% of the global drainage area, and 76% of the population lives there (Supplementary Table S7). Whereas the 'classical water scarcity' assessment based solely on water quantity indicates 984 hotspot sub-basins, covering 18% of the global land area and 42% of the global population (Table 1). Compared to the clean-water scarcity assessment, the classical assessment shows a much lower level of water scarcity in South China, Europe, and North America, where water pollution levels are high as the result of urbanization and agricultural activities, and the relatively high runoff that transports N to rivers (Fig. 2, Supplementary Figs. S4 and S8). Clearly, quality-induced water scarcity hotspots affected more ecosystems and people than quantity-induced water scarcity hotspots in 2010 (Supplementary Figs. S5 and S6 in SI).

Like hotspot regions, scarcity of clean water in non-hotspot sub-basins where clean-water scarcity is estimated at low or moderate levels was also dominated by nitrogen pollution. In 2010, more than 2000 sub-basins face potential water scarcity due to moderate water pollution, distributed mainly in South America and Africa, but also in North America, Northern Asia, and Australia (Fig. 2). These sub-basins cover 39% of the global drainage area and 18% of the population lives there. Sub-basins that currently do not face water scarcity issues with high water quality and high water quantity are mostly in sparsely populated regions such as northern parts of North America and Asia, and central parts of Australia (Fig. 2).

In 2050, the number of clean-water scarcity hotspot sub-basins is projected to remain high (2587 in SSP1-RCP2p6, 2785 in SSP2-RCP2p6) or even increase (3061 in SSP5-RCP8p5), with quality-induced scarcity dominating the globe (Figs. 2 and 3). In the worst-case scenario (SSP5-RCP8p5), the clean-water scarcity hotspots are calculated to cover 48% of the total drainage area, compared to 32% in 2010, and with 91% of the total global population living there, compared to 80% in 2010 (Fig. 2). Like in 2010, water scarcity hotspots are mostly found in sub-basins with intensive agricultural activities (Fig. 4). In 2050, the shares of agricultural area, nitrogen inputs, and surpluses in the hotspot sub-basins to the global total have large increases among the scenarios. For example, in SSP5-RCP8p5, the hotspots regions cover 68% (44% in 2010) of the global agricultural land. These regions are projected to receive 89% and 80% of the global nitrogen inputs from fertilizer and manure, respectively, in 2050 (this was 84% for fertilizers and 53% for manure in 2010). They are projected to contribute to 84% of the global

**Table 1 | The shares (% of the global total) of area (sub-basins drainage area), population, N (nitrogen) losses to rivers from human waste, agriculture land, N fertilizer application in agriculture, N manure application in agriculture, N in harvested crops, and N surplus in agriculture (defined as total N inputs to agriculture minus N outputs by crop uptake and animal grazing) in the 'Classical water scarcity' (WS) and 'Clean-water scarcity' (CWS) in 2010 and 2050**

| Year | % of the global total | | | | | | | | | | | | | | | |
|---|---|---|---|---|---|---|---|---|---|---|---|---|---|---|---|---|
| | Area | | Population | | N Human Waste | | Agricultural land | | N fertilizer | | N manure | | N Harvested Crop | | N surplus in agriculture | |
| | WS | CWS | WS | CWS | WS | CWS | WS | CWS | WS | CWS | WS | CWS | WS | CWS | WS | CWS |
| 2010 | 18 | 32 | 42 | 80 | 44 | 84 | 26 | 44 | 39 | 84 | 18 | 53 | 28 | 69 | 32 | 69 |
| SSP1-RCP2p6 | 19 | 33 | 45 | 80 | 49 | 82 | 27 | 47 | 34 | 79 | 25 | 64 | 31 | 69 | 32 | 70 |
| SSP2-RCP2p6 | 19 | 37 | 47 | 84 | 50 | 85 | 27 | 53 | 36 | 83 | 26 | 70 | 31 | 75 | 33 | 74 |
| SSP5-RCP8p5 | 19 | 48 | 46 | 91 | 49 | 91 | 28 | 68 | 33 | 89 | 25 | 80 | 32 | 84 | 31 | 84 |

Clean-water scarcity hotspots are sub-basins where either the levels of scarcity for water quantity or quality or both are considered high in Table 2. For 2050, clean-water scarcity is calculated for three scenarios: SSP1-RCP2p6, SSP2-RCP2p6, SSP5-RCP8p5. Details of the scenarios based on the Shared-economic pathways (SSPs) and Representative Concentration Pathways (RCPs) are available in Supplementary Tables S9–S11 in the Supporting Information.

**Table 2 | Thresholds and matrix for clean-water (quantity- and quality-based) scarcity**

| Clean-Water Scarcity | | Quantity-based ($S_{quantity}$) | | |
|---|---|---|---|---|
| | | Low (0–0.2) | Moderate (0.2–0.4) | High (>0.4) |
| Quality-based ($S_{quality}$) | Low (0–0.45) | | | Hotspot |
| | Moderate (0.45–1) | | | Hotspot |
| | High (>1) | Hotspot | Hotspot | Hotspot |

The rows and columns with High quantity-based, quality-based, or both quantity- and quality-based water scarcities are identified as hotspots of clean-water scarcity. More information of how the thresholds are derived is available in Section Hotspot analysis in Methods.

agricultural N surplus in 2050 (this was 69% in 2010). Another important contributor to water pollution in the hotspot sub-basins in SSP5-RCP8p5 is the N losses to rivers from human waste. These N losses are projected to account for 91% of the total global losses due to the high sewage connections and poor wastewater treatment in this economic-first and highly urbanized future scenario.

The spatial distribution of the clean-water scarcity hotspots in 2050 is similar to 2010, but it is estimated to expand to many continents, mostly in Africa (Figs. 3 and 4). Africa is projected to have large increases in water scarcity, mainly caused by severe water pollution. In 2010, water scarcity hotspots in Africa cover 20% of the continental area, with 27% of the continental population living in the African sub-basins in 2010. This is projected to increase to 66% (drainage area) and 88% (population) if no improved water management options are adopted in 2050 (SSP5-RCP8p5). Yet even with improved water management (SSP1), we still project an increase to 27% (drainage area) and 41% (population) in the hotspots. The water scarcity issues will remain severe in other continents, particularly in Asia, Central America, Europe, and North America.

### Water pollution is an important cause of water scarcity

Our water scarcity assessment shows that nitrogen pollution in rivers is an important cause of water scarcity in 2010 and will likely continue causing water scarcity in 2050. This calls for urgent proactive pollution control strategies to reduce the impact of future potential water scarcity on nature and humans. A better understanding of the spatial distribution and main sources of nitrogen pollution to develop such strategies is needed. We, therefore, use the MARINA-Nutrients (Model to Assess River Inputs of pollutaNts to seAs) model and analyze nitrogen inputs to rivers at the sub-basin scale by sources for 2010 and 2050.

In 2010, we estimate a total amount of 106 Tg/year total dissolved nitrogen (TDN) inputs to rivers, with the highest loads occurring in China, India, Central Africa, South America, and parts of North America (Fig. 5 and Supplementary Fig. S7). In 2050, the total TDN inputs to rivers are expected to be 112–147 kton/year among the three scenarios. This corresponds to an increase of 6–39% compared to 2010. Taking the worst-case scenario (SSP5-RCP8p5), this increase can be explained by the increased anthropogenic sources such as human waste and synthetic fertilizers (Fig. 5). In this scenario, sewage is projected to become the dominant source of nitrogen pollution in rivers mainly due to the activities around fast urbanization (i.e., population growth, more population connected to sewage systems in cities), and insufficient wastewater treatment technologies and infrastructures in this scenario. The dominant source of nitrogen pollution in the SSP1-RCP2p6 and SSP2-RCP2p6 scenarios differ from SSP5-RCP8p5. Agriculture (i.e. fertilizer application) is the most important source in these two scenarios as the result of food production activities to feed the growing population. Sewage has much smaller contributions in these two scenarios, benefiting from the improved sewage connection as well as improved treatment. The highest TDN inputs in SSP1-RCP2p6 and SSP2-RCP2p6 are found in similar regions as described for 2010 (Fig. 5).

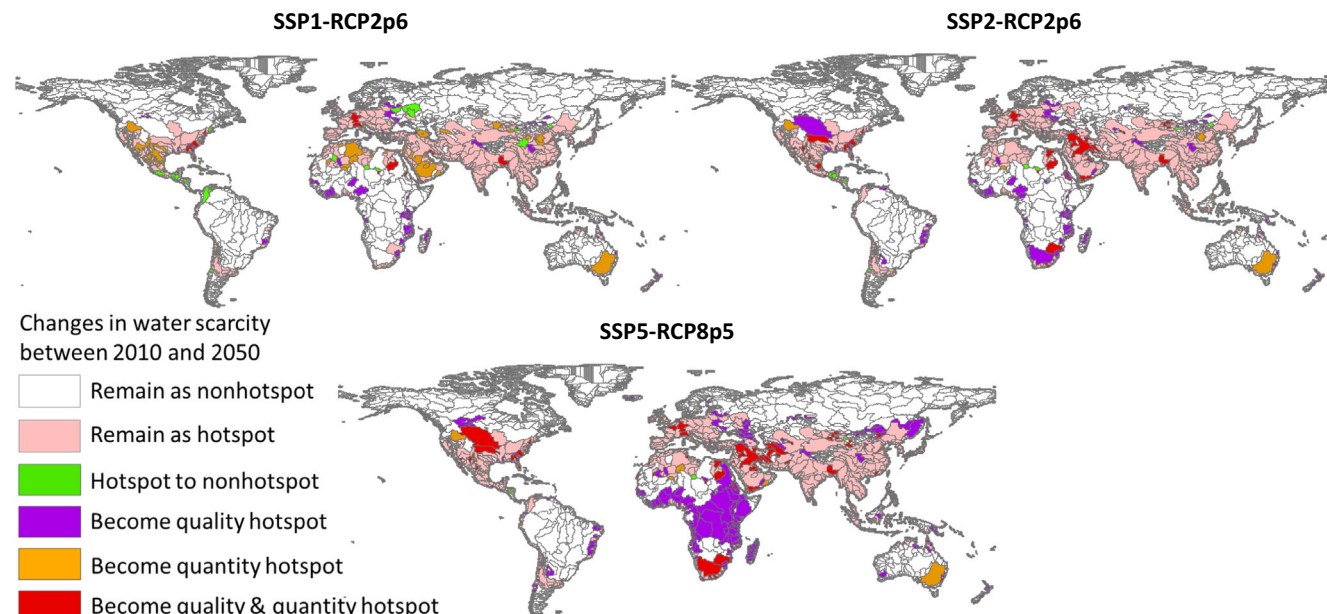

**Fig. 3 | Changes in clean-water scarcity between 2010 and 2050.** The changes are projected for three scenarios: SSP1-RCP2p6, SSP2-RCP2p6, SSP5-RCP8p5. Details of the scenarios based on the Shared-economic pathways (SSPs) and Representative Concentration Pathways (RCPs) are available in Supplementary Tables S9–S11 in the Supporting Information. Hotspots are sub-basins where either the levels of scarcity for water quantity or quality or both are considered high in Table 2.

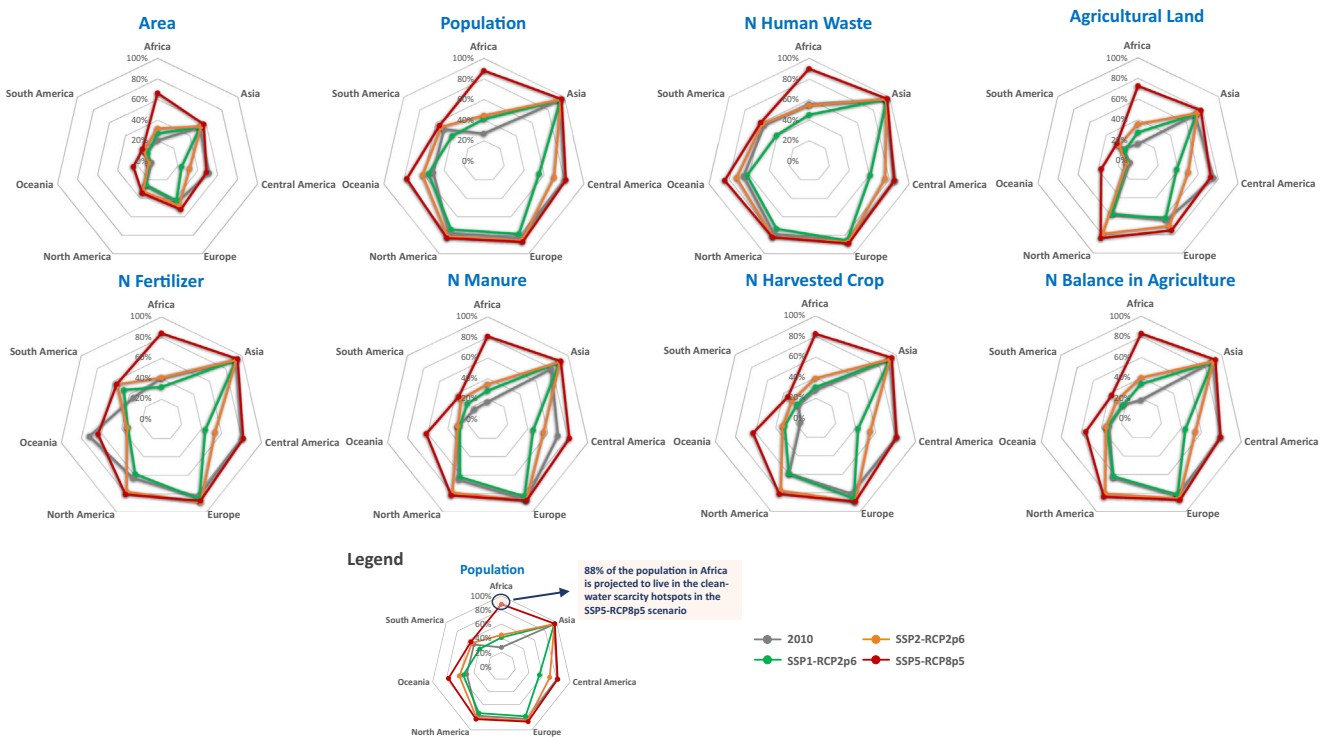

**Fig. 4 | Characteristics of the clean-water scarcity hotspots in 2010 and 2050.** The spider charts show the shares of area (sub-basin drainage area), population, N (nitrogen) inputs to rivers from human waste, agriculture land, N fertilizer application in agriculture, N manure application in agriculture, N in harvested crops, and N surplus in agriculture (defined as total N inputs to agriculture minus N outputs by crop uptake and animal grazing) in the clean-water scarcity hotspots (% of the continental total) in 2010 and 2050. Clean-water scarcity hotspots are sub-basins where either the levels of scarcity for water quantity-driven or quality-driven or both are considered high in Table 2. For 2050, clean-water scarcity is calculated for three scenarios: SSP1-RCP2p6, SSP2-RCP2p6, SSP5-RCP8p5. Details of the scenarios based on the Shared-economic pathways (SSPs) and Representative Concentration Pathways (RCPs) are available in Supplementary Tables S9–S11 in the Supporting Information.

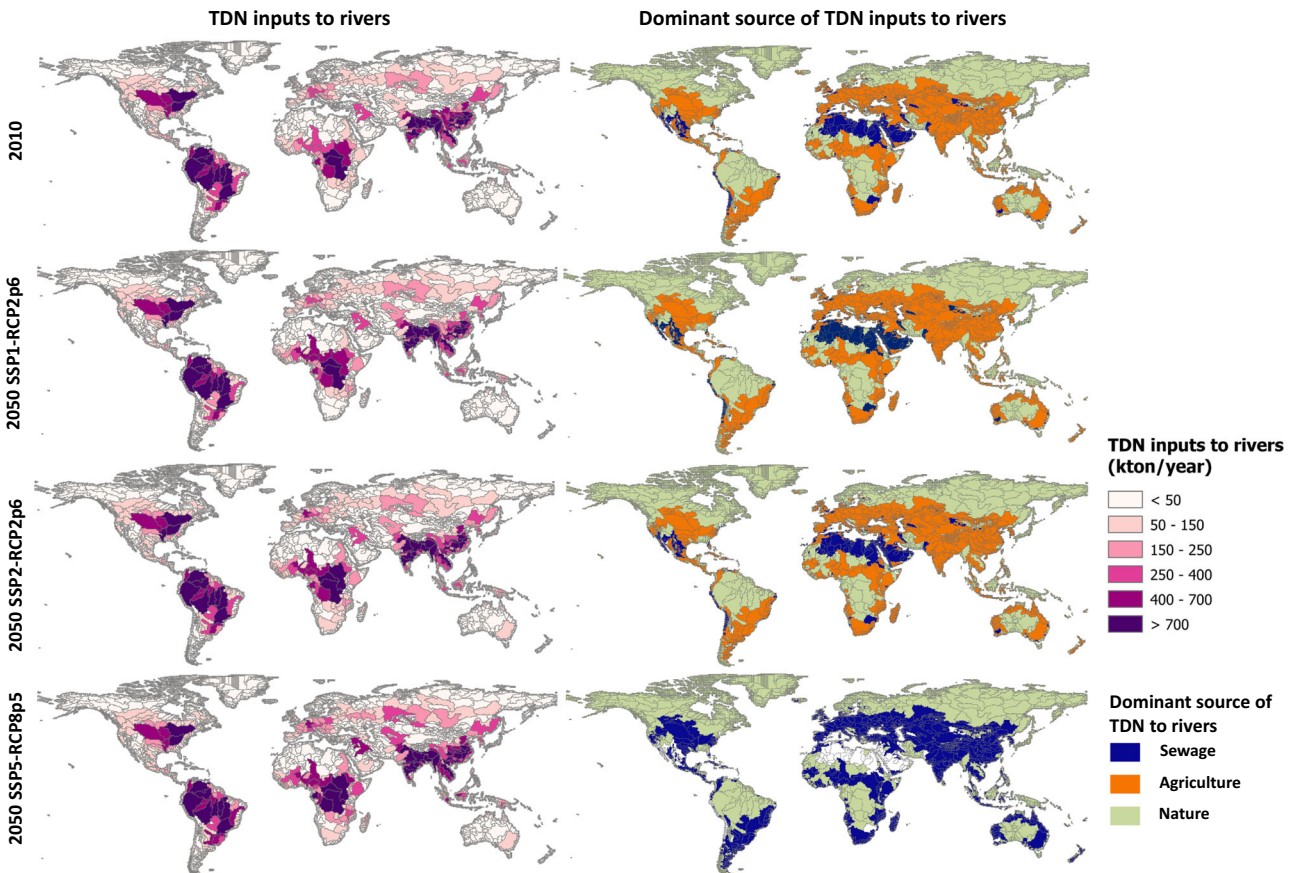

**Fig. 5 | Nitrogen inputs to rivers and their dominant source.** The maps below show Total Dissolved Nitrogen (TDN) inputs to rivers (left panel) and the dominant source of TDN inputs to rivers (right panel) at the sub-basin scale in 2010 and 2050. For 2050, three scenarios are analyzed: SSP1-RCP2p6, SSP2-RCP2p6, SSP5-RCP8p5.

Details of the scenarios based on the Shared-economic pathways (SSPs) and Representative Concentration Pathways (RCPs) are available in Supplementary Tables S9–S11 in the Supporting Information.

## Water quantity remains important

In addition to the increasing scarcity of clean water caused by water pollution, quantity-induced scarcity driven by water availability and water withdrawals remains an important issue in 2050 scenarios. This is due to the climate change-induced changes in water availability through alterations in the hydrological cycle and the increasing water demand or withdrawals driven by socio-economic developments.

Water availability is projected to increase in more than two-thirds (69–72%) of the sub-basins and decrease in the rest of the sub-basins among the scenarios between 2010 and 2050 (Supplementary Fig. S8). In our study, water availability is defined as the cumulative natural river discharges at the sub-basin outlets. However, the changes in water availability vary largely among the continents and individual sub-basins. For example, the total water availability (the sum of river discharges of sub-basins) in Africa, Asia and North America is expected to increase by 4–6% between 2010 and 2050 for the three scenarios. In contrast, the total water availability in Central America, Europe, Oceania and South America is projected to decrease by up to 4% during 2010–2050. The changes among the individual sub-basins vary from −156 to +117 km$^3$/year between 2010 and 2050, showing large extremes in water availability which may exaggerate water scarcity in dry regions in 2050. Global water withdrawals are projected to increase by 10–12% among SSP-RCP scenarios in 2050 (Supplementary Figs. S10 and S11 in SI), leaving future generations under increasing pressure of water scarcity. This increase is due to the future higher water demand, mostly from urbanization and food production.

## Different challenges among hotpots

While future hotspots of clean-water scarcity are identified mainly in China, India, Europe, North America and in the worst-case scenario (SSP5-RCP8p5) also in Central Africa. The causes of water scarcity differ among these regions, presenting different challenges that need to be addressed to reduce water scarcity.

For quantity-induced scarcity, the main causes are the excessive withdrawals (high water withdrawal over water availability). The share of water withdrawals among sectors varies largely across continents (Supplementary Fig. S12 in SI). Irrigation contributes most to surface water withdrawals on the global scale and is the most important driver of quantity-induced scarcity in most regions including China, India and South America. However, this differs in Europe, where irrigation contributes to less than 30% of water withdrawals. The most important water withdrawal is the industrial sector in Europe. A similar situation is observed in North America, where the industry takes almost 50% of the continental water withdrawal due to the large water demand for energy production (e.g., thermoelectric power plant) and manufacturing[33].

For quality-induced scarcity, the main causes of high TDN inputs to rivers are also different among hotspots. In 2010, TDN inputs to rivers are mainly driven by the low nitrogen use efficiencies in China and India (Supplementary Table S3 in SI), high production in Europe and North America (Supplementary Fig. S19 in SI), and by atmospheric N deposition and fixation on natural land in South America and Central Africa. In the future, the main cause of TDN inputs to rivers is similar across most hotspots in SSP1 and SSP2, which is agricultural production. It is important to note that although the nitrogen use efficiencies

have improved to high levels, the high food production in China, India, Europe and North America (e.g., Mississippi river) driven by food demand still leads to high N surpluses in agriculture (Supplementary Fig. S20 in SI). In SSP5, pollution is driven worldwide mainly by sewage as described above due to global urbanization and inadequate development of sewage treatment. Atmospheric N deposition and fixation on natural land remain the main source of TDN inputs to rivers in South America and Central Africa in the future, while agricultural N surpluses become increasingly more important in SSP5 (Supplementary Fig. S20 in SI).

## Discussion

There is a growing awareness that water quality aggregates water scarcity in many regions. Nonetheless, most global water scarcity studies focused solely on water quantity aspects[6]. To our knowledge, there are two indicators that consider water quality explicitly. The first is the water quality dilution (WSq) indicator[28,29], which has been applied for several water quality constituents (e.g. temperature, salinity, organic pollution, total nitrogen and total phosphorus) on a global scale. The second is the Quantity-Quality-Environmental flow requirement (QQE) indicator[26,27], which has only been applied in China. Based on these indicators, we introduce 'clean-water scarcity' as a terminology and present a global assessment accounting for both water quantity and quality based on global nitrogen pollution under different scenarios of climatic and socio-economic developments. To do this, we developed and applied two water scarcity indicators (a water quantity-based and quality-based indicator) for global rivers at the sub-basin scale.

Overall, our assessment identifies similar hotspot regions for 2010 as those existing studies based only on the water quantity aspects of water scarcity in the period of 1971–2005[1,7,13,34,35]. However, our scenario analysis reveals a higher number of water scarcity hotspots than the existing quantity-focused studies. Moreover, our assessment provides a more detailed view on non-hotspot areas by differentiating between low, moderate, and high levels of scarcity, considering both quantity and quality aspects. Such information has the added value of identifying those regions facing potential water scarcity challenges if water resources and pollution are poorly managed. For example, the baseline simulation (1971–2000) in the study of Hanasaki et al.[13] identifies similar water scarcity regions as we do for 2010. Whereas their study predicts reduced future water scarcity in Central Europe, India, China, and Africa, we find increased scarcity in those regions because of poor water quality. While the use of different models and datasets in the above-mentioned studies affects the distribution of water scarcity regions, the differences originate largely from neglecting the impact of water pollution (nitrogen in our study) on water scarcity.

Estimations of the population experiencing scarcity of clean water in our study are generally higher than in previous studies focusing only on water quantity[3,4,7,36]. Previous studies estimated that between 25% and 65% of the population lived in severely water-scarce areas in 1995–2005. Looking only at water quantity, we estimated that 45% of the global population lived in areas with severe water scarcity in 2010. Yet, if we also take into account water quality issues, our estimate rises to 80% of the global population living in water-scarce regions in the same year. This strongly agrees with the study of Vörösmarty et al.[37] who found that nearly 80% of the world's population lives in areas facing water security challenges from both water quantity and quality perspectives.

We consider quality-based water scarcity as severe if nitrogen concentrations exceed the threshold (1 mg/L TDN) to avoid eutrophication in aquatic ecosystems, following De Vries et al.[38], Yu et al.[39]. This may lead to a bias toward environmental water scarcity. Another way to deal with the different thresholds could be to focus on N thresholds for specific sectoral uses rather than the total water availability in line

with the study of van Vliet et al.[29], who identified a similar global spatial distribution of clean-water scarcity as we identify. For example, the threshold for the agricultural sector is linked to irrigation withdrawals (80 mg/L N), while the threshold for drinking water (11 mg/L N) is used to calculate domestic withdrawals. Water scarcity could be assessed for each sector specifically instead of computing a total water scarcity. However, water allocation-related issues arise here, such as how to include water allocation among sectors. Here we decided to take the stringent threshold for the aquatic ecosystem whose ecosystem functions (e.g., irrigation water supply, fishery, recreation) will negatively impact both nature and humans. While one can argue that this strict threshold may lead to an overestimation of quality-based water scarcity, even stricter standards (0.25–0.5 mg N/L for rivers, comparable to the moderate levels in our assessment) have been introduced by United Nations to assess water security for ecosystem and human use in their SDG guidelines[40].

We conducted a clean-water scarcity analysis on annual temporal and sub-basin spatial scales with a focus on nitrogen pollution. The assumption is that the annual cumulative natural river discharge at sub-basin outlets represents water availability, annual summed gridded sectoral withdrawals represent water withdrawals, and the annual cumulative N load represents water quality in the sub-basins. Consequently, the water scarcity indicators generate an average value for each sub-basin, which aggregates the differences in time and space. Yet, our assessment masks intra- and inter-annual variabilities in the freshwater resources[3,41]. Most importantly, the annual discharge does not represent the proportions of river flow that derive from base flow and stormflow. This means that the annual discharge might consist of a constant base flow available throughout the year but could also represent a high storm flow with a relatively low base flow. Whilst the former limits water scarcity, the latter aggravates among seasons. This issue is apparent in the future climate change scenario, particularly in RCP8p5 regional higher water availability might originate from higher seasonal fluctuations with higher stormflows[42]. This holds the same for the EFRs in our study, for which we calculated annual EFRs as a fraction (ranging between 30–38%) of the total available water for sub-basins based the Variable Monthly Flow Method approach on Pastor et al.[43]. The intra-annual variabilities in very wet or dry rivers may lead to a higher or lower monthly EFR in these rivers, ranging from 30 to 60% among sub-basins and seasons globally. Additionally, conducting a water scarcity analysis on an annual scale does not consider seasonal variability in nitrogen pollution[44,45]. Exner-Kittridge et al.[45] have observed a seasonal trend in nitrogen concentrations in rivers that increases in winter and decreases in summer. This phenomenon can be attributed to various factors such as higher in-stream nitrogen uptake and denitrification rates during the summer compared to the winter, seasonal biochemical changes, or the seasonal timing of fertilizer application and plant uptake. Thus, whilst our results indicate overall annual water scarcity, the level of scarcity may vary largely among seasons. This means that our assessment should be used and interpreted to estimate the global distribution of clean-water scarcity and their trends over time rather than zoom into the details of intra-annual scarcity in specific regions.

We assessed clean-water scarcity based on the integrated modeling framework (Fig. 1) that links land use and agriculture, hydrological and water quality models. While the models provide great opportunities for exploring future trends and causes of water scarcity, there are uncertainties around the model inputs and modeling approaches of MAgPIE, VIC and MARINA-Nutrients. Below we discuss why we consider our modeling approach reliable and sufficient in assessing future clean-water scarcity.

Uncertainties in our assessment from MAgPIE are mainly related to the estimated nitrogen (N) budgets (see Section MAgPIE in SI for the modeling approach). To build trust in our assessment, we compared our N budget from MAgPIE with a recent high-resolution (5 arc min

degree) N dataset from Tian et al.[21]. The result shows a promising comparison. The global total nitrogen inputs to agriculture (cropland and pasture) and non-agriculture are very comparable, despite small differences among the sources as fertilizer, manure, and deposition (Supplementary Tables S8 in the SI). For example, the global total N input in 2010s is 267 Tg/year in Tian et al.[21] and 287 Tg/year in this study. The spatial distribution of N inputs is also comparable between Tian et al.[21] and our study (Supplementary Fig. S21 in SI). High total N inputs are observed in China, South Asia, Europe, United States and Brazil in both studies. There is an exception for atmospheric deposition. Some regions such as South Africa have higher deposition than in other regions in Tian et al.[21] but not in our study. However, the N load quantified by the MARINA model is not sensitive to changes in atmospheric deposition due to its small contribution to water pollution compared to other sources. This was revealed in the thorough sensitivity analysis in Appendix E of Wang et al.[46], −/+10% changes in atmospheric deposition in the MARINA model hardly result in any difference in river export of N in MARINA. Considering the comparable total N inputs on land between the two studies, we believe that our results of quality-induced water scarcity hotspots will not change much when using N data from Tian et al.[21]. This comparison provides a high confidence in using the MAgPIE model for our assessment.

VIC's simulated historical discharge and sectoral water withdrawals compared well against observed discharge[47] and reported domestic, industrial and irrigation water withdrawals[33]. However, multi-model intercomparison studies have shown that differences between hydrological models are the main source of uncertainty in discharge and irrigation water demand projections[8,48,49]. Therefore, the selection of a single model, the VIC hydrological model in our study impacts the water scarcity results. Nevertheless, we believe the model validation results show an acceptable performance of VIC for assessing water scarcity. VIC model estimates are mostly near the multi-model ensemble means for discharge[50] and irrigation[51]. Moreover, the explicit representation of the energy balance in the VIC model allows the model to comprehensively capture the impacts of radiation changes under climate change, which is highly important for, for example, snow dynamics[50].

MARINA-Nutrients-Global-1.0 was developed and applied in this study to quantify N load at the sub-basin outlets. The previous versions of this model[17,46,52] have been evaluated with a convincing performance at both global and regional scales based on the 'building trust circle' approach including 1) compare model outputs with measurements and existing studies, 2) compare spatial pattern of pollution hotspots, 3) sensitivity analysis, and 4) compare model inputs with independent datasets. The MARINA model, however, does not consider the legacy pools of N. Considering the historical N use can have an impact on N pollution in groundwater[53] and rivers[54], while the legacy effects balance out over longer time periods. If N legacy effects were considered, this would likely increase the modeled water pollution in most of the hotspot regions such as China, Europe, North America where historical N use was high, confirming our conclusion that water pollution will become an important cause of clean-water scarcity in these regions. Here, we consider that the MARINA version developed in this study provides a robust assessment of quality driven-water scarcity because of the following reasons. First, the simulation of point source pollution from wastewater has been evaluated as promising in the global study of Strokal et al.[17]. Second, for diffuse source pollution we used data from MAgPIE, which shows good performance as discussed above when compared to N budget in other studies. Third, our modeled results compare very well with other global studies that quantify historical or future N pollution in rivers. For example, our global spatial patterns of N pollution are comparable to the total N patterns for 2000–2010 shown by van Vliet et al.[29]. This is the same for the future that we estimate similar hotspots of N pollution in 2010 and 2050 compared to Beusen et al.[55].

Our assessment has important Implications for future water management and policies. Strategies to adapt to or mitigate future water scarcity are urgently needed, especially as socio-economic developments continuously increase the world's dependence on water resources. Adaptation strategies currently focus on quantity-based water scarcity, varying from water-saving irrigation techniques at the sectoral scale to water diversion or reallocation through dams at the catchment scale[31,56–58]. Mitigation strategies that reduce water pollution surely need more attention, as revealed by this study, that low water quality will be a critical or even dominant cause of water scarcity in many river basins in the future, and controlling nitrogen pollution is very challenging.

The challenge to control nitrogen pollution mainly arises from current urbanization trends and increasing food demands and waste which both contribute to additional nitrogen losses to water[46,59–64]. Even in the ambitious SSP1-RCP2p6 scenario, assuming optimistic water management, as well as diet, changes towards lower shares of animal products and food waste, the scarcity of clean water remains high in many regions due to water pollution. Based on source attribution of nutrients in our study and previous water quality assessments[17,29,63], improving nutrient management in food production and sewage connection and treatment are urgently needed in densely populated sub-basins to reduce water scarcity.

We took nitrogen pollution as the water quality indicator in our study. However, many other indicators (e.g., salinity, dissolved oxygen, biological demand, pH, temperature, and heavy metals) and newly emerging pollutants (e.g., pathogens, antibiotics, plastics, and pesticides) will likely cause severe water degradation in the future[17,18,65–67]. Research is thus needed to identify the impacts of these indicators or pollutants on future water scarcity among sectors as a joint effort of the water quality community. The advantage of our clean-water scarcity indicators is that they are not limited to specific pollutants and specific temporal or spatial scales. Therefore, the indicators can be quantified for various individual pollutants across temporal and spatial scales depending on the purpose of the assessment. Another opportunity is to combine our assessment approach with the Water Quality Index (WQI) models[68–71]. WQI models are powerful tools based on aggregation functions to convert varying water quality datasets to a single water quality index to assess the quality of the waterbody. Such models can thus help to aggregate the quality-based indicators ($S_{quality}$) for individual pollutants to a simple single indicator for the water scarcity assessment. Additionally, a better understanding of the interactions between multiple water pollutants and their sources is very useful for developing strategies to simultaneously control the pollution of multiple pollutants. For example, nutrients, antibiotics, and pathogens share the same source (manure) in animal production. Improved manure management, such as treatment and recycling, will then have synergetic effects in controlling water pollution by all three pollutants.

Moreover, addressing water quality is of significant importance for achieving several Sustainable Development Goals (SDGs)[32]. Studies show that there are many interactions (synergies and trade-offs) between SDG 6 and other SDGs, such as those goals ensuring food security (SDG 2), sustainable urbanization (SDG 11), responsible production and consumption (SDG 12), and mitigated climate change (SDG 13)[46,72–75]. Our results show similar potential synergies and trade-offs between the SDGs. For example, we found that water scarcity mostly exists in sub-basins with intensive agricultural production (SDG 2) or/and are densely populated (SDG 11), leading to high pollutant loads and high water withdrawals (SDG 12) from the relatively low water availability due to climate change-induced hydrological changes (SDG 13). Strategies such as reducing fertilizer use to control water pollution in these hotspots may lead to trade-offs that challenge food provision. Such trade-offs can be turned into synergies by applying agricultural practices (e.g., alternative varieties, fertilizing crops based

on their nutrient requirement) to improve nutrient efficiencies of crops such that the crop yield is maintained and food demand is met. It is thus essential to consider the interactions between the above-mentioned SDGs in the water scarcity strategies to avoid negative impacts on achieving the goals for food, cities, and climate. Our results about the clean-water scarcity hotspots and their socio-economic and climate characteristics provide a very valuable indication of where and what interactions need to be addressed to mitigate water scarcity as well as ensure sustainable development for other domains of society.

## Methods
### Clean-water scarcity assessment
Many water scarcity indicators consider water quantity, yet only a few consider both water quantity and water quality aspects[6,26,28,29,76,77]. For example, the Quantity-Quality-EFR (QQE) indicator in Liu et al.[26] was the first being developed and applied to assess both quality and quantity-based water scarcity based on the blue water footprint, gray water footprint, and environmental flow requirements (EFRs). Another example is the water quality dilution (WSq) indicator developed and applied globally to assess historical water scarcity as a proportion of sector-specific water withdrawals of suitable water quality to the total water availability[28,29,77]. In this study we developed and applied clean-water scarcity indicators considering both aspects, inspired by the Quantity-Quality-EFR (QQE) indicator which has an advantage in quantifying the quality and quantity-based water scarcity separately. Therefore, these clean-water scarcity indicators will enhance the understanding of overall water scarcity, aiding in the determination of whether the scarcity is predominantly due to issues of quantity or quality.

### Indicators for clean-water scarcity
We assessed the scarcity of clean water for >10,000 sub-basins worldwide based on two indicators: a quantity-based indicator ($S_{quantity}$, Eq. (1)) and a quality-based indicator ($S_{quality}$, Eq. (2)).

$$S_{quantity} = \frac{\sum_{j=1}^{n}(D_j)}{Q_{nat} - \text{EFR}} \quad (1)$$

$S_{quantity}$ is the quantity-based indicator, calculated based on the criticality ratio, i.e., the rate of water use to water availability[78]. $D_j$ stands for the water withdrawals for sector j in the sub-basins (km³/year), $Q_{nat}$ for the natural river discharge at the sub-basins outlets which stands for total water availability (km³/year), and EFR for the environmental flow requirements in the sub-basins (km³/year).

$$S_{quality} = \frac{L}{Q_{act} * C_{max}} \quad (2)$$

$S_{quality}$ is the quality-based indicator. L stands for the pollutant (nitrogen in our study) load at the sub-basins outlets (kton/year), $Q_{act}$ for the actual river discharge at the sub-basins outlets (km³/year), $C_{max}$ for the maximum water quality threshold of the pollutant (mg/L) for specific purposes of water use (1 mg/L total dissolved nitrogen the threshold for sustaining the aquatic ecosystem from eutrophication[38,39]). See Section Data for clean-water scarcity assessment in Supporting Information (SI) for a detailed description of the data that we derived for the above variables.

### Modeling framework to assess clean-water scarcity
To calculate the indicators for clean-water scarcity (Eqs. (1) and (2)), we combined the MARINA-Nutrients (Model to Assess River Inputs of pollutaNts to seAs), MAgPIE (Model of Agricultural Production and its Impact on the Environment), and VIC (Variable Infiltration Capacity) models into an integrated modeling framework (in Fig. 1). We used modeled results of this modeling framework to calculate the quantity-based ($S_{quantity}$) and quality-based ($S_{quality}$) clean-water scarcity indicators. We did this for 2010 and 2050 under the Nitrogen futures in the Shared Socio-economic Pathways[79] developed based on the Shared-economic pathways (SSPs) and Representative Concentration Pathways (RCPs). The $S_{quantity}$ and $S_{quality}$ indicators were calculated for the outlets of the sub-basins to assess the level of water scarcity in rivers. Supplementary Table S1 in the SI presents in detail how each variable in Eqs. (1) and (2) was derived using the models in Fig. 1 for 2010 and 2050.

Quantity-based indicator ($S_{quantity}$) was calculated mainly based on simulations of river discharge and sectoral water withdrawal by the VIC model (Fig. 1). VIC is a macro-scale grid-based hydrological model that simulates water balance and surface energy balance (e.g., interception, evapotranspiration, surface and subsurface runoff, and river discharges) and anthropogenic water use. Here we took domestic, industrial, livestock and irrigation water withdrawals ($D_j$) in 2010 derived from VIC by the study of Droppers et al.[33]. Future sectoral water withdrawals in 2050 were derived based on the changes in water withdrawals between 2010 and 2050 from the MARINA-Nutrients-Global-1.0 model developed in this study and water withdrawals in 2010 from VIC as derived above. Natural river discharge ($Q_{nat}$) was based on the cumulative natural river discharge, i.e., discharge before water withdrawals, at the sub-basins outlets, simulated by VIC[47]. Following the approach of Pastor et al.[43], we calculated the sub-basin specific EFR for the environment, withheld from human usage to keep ecosystems in a fair ecological condition. More details on the variables ($D_j$, $Q_{nat}$, and $EFR$) used to assess $S_{quantity}$ are available in Supplementary Table S1 in Supplementary Information (SI).

Quality-based indicator ($S_{quality}$) was calculated from the nitrogen pollution perspective based on VIC, MAgPIE and MARINA-Nutrients (Fig. 1). Actual water availability ($Q_{act}$) for $S_{quality}$ was based on the cumulative actual river discharge, i.e., discharge after water withdrawals within the sub-basins, at the sub-basins outlets, derived from MARINA-Nutrients that incorporated hydrology from VIC (see Supplementary Table S1 in SI for details). For $C_{max}$, we took 1 TDN mg/L as the threshold for sustaining the aquatic ecosystem from eutrophication, based on the study of De Vries et al.[38], Yu et al.[39]. Here, L is the total dissolved N (TDN) load at the sub-basins outlets (kton/year). TDN is the sum of dissolved inorganic (DIN) and organic nitrogen (DON). DIN and DON loads at the sub-basin outlets are simulated separately by linking MAgPIE and MARINA-Nutrients (referred as $OT_{F,y,j}$ in Eq. (S3) for individual rivers or tributaries, and $OC_{F,y,j}$ in Eq. (S4) for main channel; see Supplementary Fig. S1 for the definition of tributary and main channel).

MAgPIE is a global land-system modeling framework[64,80] that simulates long-term scenarios for the global land and food system. It is a recursively dynamic model that simulates how food, feed and material demand can be fulfilled under different possible future pathways. The model estimates the extent and distribution of agricultural land (cropland and pastureland), forest areas and other natural lands for the future until the year 2100. MAgPIE estimates nitrogen budgets on the level of 18 global world regions, which are downscaled to 0.5° grid level in the model post-processing and used as inputs for MARINA-Nutrients to simulate nitrogen pollution from diffuse sources. A detailed description of MAgPIE is available in Section MAgPIE in SI.

MARINA quantifies the annual river export of multiple pollutants (i.e. nitrogen (N), phosphorus, micro- and macro plastic, pathogens, and chemicals) to seas from point and diffuse sources for >10,000 subbasins worldwide[65]. In our study, we developed the MARINA-Nutrients-Global-1.0 model to assess river and coastal water pollution by total dissolved N (TDN) to seas from both diffuse and point sources in 2010 and 2050. TDN includes dissolved inorganic and organic N (DIN and DON). N from diffuse sources in agriculture and non-agriculture land is based on MAgPIE, while N from point

sources is based on MARINA-Multi-Global-1.0 developed by Strokal et al.[17] to quantify nitrogen inputs to global rivers from sewage systems (Fig. 1). Details in model equations and model inputs for MARINA-Nutrients are available in Section MARINA in SI.

While developing this modeling framework, we also acknowledge that there are many other very useful models which could be adopted to assess clean-water scarcity. Many global hydrological models exist to assess water availability[47,50,81]. The land use and agriculture model IMAGE, for example, is another well-known model in the field for simulating land use and nitrogen budget[82]. Moreover, many water quality models exist to assess water pollution from various pollutants across temporal (e.g. daily, seasonal, annual) and spatial scales (e.g., catchment, sub-basin, basin). For example, SWAT[83], WorldQual[82,84], and RTM[85] and IMAGE-GNM[55] have been proven to be advanced models for assessing N pollution in rivers. In this study, we decided to use the MARINA-Nutrients-Global-1.0 model as the starting point for clean-water assessment, in combination with MAgPIE and VIC, because of the following strengths. First, the MARINA model allows quantifying water pollution by source (i.e., fertilizer, manure, sewage and etc) which helps to better understand the main causes of water pollution. Second, the sub-basin approach of MARINA provides an opportunity to analyze water scarcity in large river basins in more detail. Moreover, not all models are available to assess future trends in clean-water scarcity under the SSPs and RCPs, while the combination of MAgPIE, VIC and MARINA is capable for such future assessment.

## Hotspot analysis

Based on the quantity- and quality-based indicators, we identified the global hotspots for clean-water scarcity. First, we determined the water scarcity levels for the sub-basins by low, moderate, and severe levels for both quantity and quality (Table 2). For quantity-based water scarcity ($S_{quantity}$, Eq. (1)), we used the generally accepted thresholds of >0.2 for moderate water scarcity and >0.4 for severe water scarcity. These levels were derived from existing studies[6,29,86,87]. For example, quantity-based water scarcity is considered severe when more than 40% of the available water – available river discharge after subtracting the amount for sustaining the environmental flow requirement - is used ($S_{quantity} > 0.4$)[29]. For quality-based water scarcity ($S_{quality}$, Eq. (2)), we used the thresholds of >0.45 for moderate water pollution and >1 for severe water pollution in view of avoiding eutrophication in aquatic ecosystems. The threshold of >0.45 means that TDN concentrations at the outlets of the sub-basins are higher than 0.45 mg N/L, indicating that the surface water bodies start to switch from the oligotrophic (clear water) to mesotrophic or eutrophic (turbid water) states, according to the trophic state index (TSI)[88]. The threshold of >1 means that nitrogen concentrations in the water bodies are higher than 1.0 mg N/L which can result in eutrophication in aquatic ecosystems[38,39]. Next, we identified the hotspots as the sub-basins where there is a severe quantity-based, quality-based, or both quantity- and quality-based water scarcity (see rows and columns with High in Table 2).

## Scenarios

We assessed water scarcity for the current (2010) and future (2050) years. For 2050, three scenarios along the storylines of the Nitrogen futures in the Shared Socio-economic Pathways[79] and storylines of future urbanization and wastewater management[17] developed based on the Shared-economic pathways (SSPs) and Representative Concentration Pathways (RCPs) were applied. These scenarios are: SSP1-RCP2p6, SSP2-RCP2p6, and SSP5-RCP8p5 (Supplementary Table S9 in SI). SSP1-RCP2p6 assumes a future focusing on sustainable socio-economic development, high-ambition N policies and an ambitious diet shift to a low meat diet, improved sewage connection and treatment, sustainable water withdrawal, combined with strong climate mitigation and its impacts on hydrology. SSP2-RCP2p6 assumes a socio-economic development following the historical trends, moderate-ambition N

policies and medium meat & dairy diet, not much-improved sewage connection and treatment, not much-changes in water withdrawal, combined with strong climate mitigation and its impacts on hydrology. SSP5-RCP8p5 assumes an urbanized future with fossil-fuel-driven socio-economic development, low-ambition N policies and meat & dairy-rich diet, improved sewage connection but limited improvements in sewage treatment, high water withdrawal, combine with low climate mitigation and its impacts on hydrology. The scenario assumptions for sewage connection and treatment are available in Supplementary Table S10 in SI and in Strokal et al.[17], which are implemented in MARINA-Nutrients-Global-1.0 for modeling nutrient pollution in rivers from point sources. The scenario assumptions for land use and agriculture are described in further detail in Supplementary Table S11 in SI, which is implemented in the MAgPIE model to produce model inputs for MARINA-Nutrients-Global-1.0 for modeling nutrient pollution in rivers from diffuse sources.

## Data availability

All data of the clean-water scarcity assessment newly generated and analyzed in this study are publicly available in the Data Archiving and Networked Services (DANS Easy) repository https://doi.org/10.17026/PT/3ICWZM.

## Code availability

All equations for the MARINA model are provided in the Supplementary information files of the following open access publication: Wang, M., Kroeze, C., Strokal, M., van Vliet, M.T.H. & Ma, L. Global change can make coastal eutrophication control in China more difficult. Earth's Future 8, e2019EF001280, https://doi.org/10.1029/2019ef001280 (2020).

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

## Acknowledgements

We acknowledge the support of the KNAW-MOST project: Sustainable Resource Management for Adequate and Safe Food Provision (SURE+) (PSA-SA-E-01, supporting M.W.), the Dutch Talent Program Veni-NWO project (0.16.Veni.198.001, supporting M.S.). We acknowledge the support of Food, Agriculture, Biodiversity, Land-Use, and Energy (FABLE) Consortium (FABLE 2.0, Grant 94120, supporting F.B.), European Union's Horizon 2020 research and innovation program Grants (776479 COACCH and 821010 CASCADES, supporting B.L.B.) and German Ministry for Education and Research (BMBF) Grant (01LS2105A, ABCDR, supporting B.L.B.). We also acknowledge the European Union (ERC Starting Grant, B-WEX, Project 101039426, supporting MTHvV) and Netherlands Scientific Organisation (NWO) VIDI grant (VI.Vidi.193.019, supporting MTHvV).

## Author contributions

M.W. and R.R. designed the research. M.W., B.L.B., F.B., R.R., M.P.B., M.B. and A.P. performed the modeling assessment of clean water scarcity, which were reviewed and commented on by B.D., M.T.H.v.V. and M.S. M.W. and M.T.H.v.V. performed the quantification of environmental flow requirements. M.W. drafted the first version of the manuscript. All co-authors contributed to the interpretation of the results, critical revision of the manuscript, and approval of the final version of the manuscript.

## Competing interests

The authors declare no competing interests.

## Additional information

**Peer review information** : *Nature Communications* thanks Junguo Liu, Md Galal Uddin and the other, anonymous, reviewer(s) for their contribution to the peer review of this work. A peer review file is available.

