## [Peer Review File · Nature Communications]

REVIEWER COMMENTS

Reviewer #1 (Remarks to the Author):

The concept of the manuscript is very interesting and has scientific value. However, I have a few queries regarding some technical aspects. For detailed comments and recommendations, please add them in the MS-Word review pane.

Reviewer #2 (Remarks to the Author):

In the backdrop of the ever-evolving global landscape, water scarcity has emerged as a formidable impediment to the attainment of the Sustainable Development Goals (SDGs) set forth by the United Nations. Within the framework of SDG Target 6.4, it is explicitly stipulated that, "By 2030, there should be a substantial enhancement in water-use efficiency across all sectors, coupled with the establishment of sustainable practices for the extraction and provision of freshwater. These measures aim to address the pressing issue of water scarcity and significantly alleviate the plight of those afflicted by its repercussions." Although numerous studies have been dedicated to investigating water scarcity since the 1980s, the incorporation of water quality into water scarcity assessments remained a rarity until the advent of the 2010s. This research void was aptly recognized by the pioneering work of Junguo Liu and his associates, who spearheaded the integration of water quantity, water quality, and environmental flows in water scarcity assessments. Inspired by their groundbreaking contributions, several studies have emerged at various spatial scales, exemplified by the comprehensive global analysis by van Vliet et al. (2017) published in *Nature Geoscience* (10: 800–802), and the focused examination at the national level for China conducted by Ma et al. (2021) in *Nature Communications* (11: 650). Despite the commendable strides made thus far, a notable gap remains in terms of future water scarcity projections that effectively amalgamate quantity-based and quality-based assessments.

This paper unveils a captivating exploration into the realm of global clean water scarcity, encompassing a comprehensive analysis of water quantity, water quality, and environmental flows. Notably, this study presents a distinctive advancement by delivering a holistic assessment, encompassing both historical and prospective perspectives, of quantity- and quality-induced water scarcity across more than 10,000 sub-basins worldwide. A noteworthy revelation emerges from the research, emphasizing the imperative to shift our focus towards quality-induced water scarcity, as the findings reveal a threefold surge in basins grappling with future pollution-induced water scarcity on a global scale. This significant discovery underscores the urgency of devoting heightened attention to the intricate interplay between water quality and the alarming issue of water scarcity.

This study presents a compelling and captivating analysis, demonstrating meticulous attention to detail and a thoughtful examination of the subject matter. The manuscript exhibits a commendable structure that ensures readability and accessibility for readers. However, it is important to note that a few concerns warrant further consideration and should be diligently addressed in a revised version of the paper.

Abstract:

In lines 22-24, the comparison between different assessments, with and without consideration of quality-induced factors, lacks clarity. It only becomes apparent after reading the entire manuscript. To improve clarity, please provide a more direct explanation of this comparison.

Introduction:

In lines 35-40, during the discussion on the impacts of climate change on the hydrological cycle and water resources, it is important to acknowledge a significant literature reference. The Intergovernmental Panel on Climate Change (IPCC) has released a report (WGII) specifically dedicated to addressing water-related issues. I strongly recommend that the authors thoroughly study this report and integrate relevant research findings and advancements into their discussion. This inclusion will enrich the analysis and provide a comprehensive overview of the current state of knowledge on this topic.

Caretta, M.A. et al., 2022. Water. In: *Climate Change 2022: Impacts, Adaptation, and Vulnerability. Contribution of Working Group II to the Sixth Assessment Report of the Intergovernmental Panel on Climate Change* [H.-O. Pörtner, D.C. Roberts, M. Tignor, E.S. Poloczanska, K. Mintenbeck, A. Alegría, M. Craig, S. Langsdorf, S. Löschke, V. Möller, A. Okem, B. Rama (eds.)]. Cambridge University Press, Cambridge, UK and New York, NY, USA, pp. 551-712, doi:10.1017/9781009325844.006.

Line 50-52: It is recommended to provide references to support these statements to reinforce the validity of the claims made.

Line 52-57: This section lacks key references, including an early study published in 2010, which investigated spatial patterns of 11 nitrogen inputs/outputs on a global scale with high spatial resolution. Additionally, a recent publication by Tian et al. (2022) on high-resolution nitrogen inputs is also missing. It is crucial to include these references to ensure a comprehensive analysis of the topic.

Moreover, it would be beneficial to compare the results obtained in the present study with those of Tian's study. This comparison will allow the authors to highlight the advantages of the nitrogen method employed in their research in contrast to Tian's approach.

Liu J., You L.Z., Amini M., Obersteiner M., Herrero M., Zehnder A.J.B., Yang H. 2010. A high-resolution assessment on global nitrogen flows in cropland. *Proceedings of the National Academy of Sciences of the United States of America* 107(17): 8035-8040.

Tian H., et al., 2022. History of anthropogenic Nitrogen inputs (HaNi) to the terrestrial biosphere: a 5-arcmin resolution annual dataset from 1860 to 2019. *Earth System Science Data* 14 (10): 4551-4568.

Line 58-63: There is an opportunity to enhance the presentation of research progress in this section. While references 21-22 focus on regional studies, it is worth noting that they are pioneering work on water scarcity assessment with consideration for water quality. By conducting a thorough investigation of subsequent studies, it becomes evident that most global-scale studies align with the fundamental principles outlined in references 21-22, with similar methodologies being employed.

By highlighting these aspects, the authors can emphasize the foundational contributions of references 21-22 and demonstrate the consistency in approaches taken by subsequent studies on a global scale.

Results

In line 88, it is unnecessary to repeat the definition of water scarcity, as it has already been provided earlier in the manuscript.

Lines 159-171: While the paper emphasizes the global changing trends in water scarcity, it would also be valuable to explore the specific trends in different countries, such as China, India, and the US. Understanding the variations and patterns of water scarcity in these countries can provide further insights into regional dynamics and help tailor localized strategies and interventions. Therefore, it is recommended to consider including a discussion on the trends and implications of water scarcity in specific countries of interest, along with their unique challenges and potential solutions.

Line 185: 4355%? How come?

Discussion

Line 205-205: The statements made in this section are not entirely accurate. "Clean water scarcity" is not a new concept but rather a recent terminology used to emphasize the importance of addressing both the quantity and quality aspects of water scarcity.

Line 261-262 and other related sections on environmental flow requirements: It is crucial to recognize that different river basins have varying levels of environmental flow requirements. Several studies have been conducted on this topic, providing valuable insights into the specific needs of individual river basins. To enhance the accuracy and effectiveness of the assessment, it is advisable to incorporate state-of-the-art findings on environmental flow requirements. By considering these findings, the study can

avoid applying a single value for all situations, such as the mentioned 37%, and instead adopt a more nuanced approach that accounts for the unique characteristics and ecological demands of each river basin.

Line 284-285: When discussing food waste and losses, it is important to provide references to support the statements made. Over the past years, there have been significant research efforts dedicated to this topic, resulting in a wealth of literature on the subject. By incorporating relevant references, the paper can strengthen the credibility of its discussion on food waste and losses and ensure that the information presented is backed by established research in the field.

Method/Discussion

It would be beneficial for the authors to discuss the uncertainties associated with the models utilized in their study, namely VIC for water and MARINA-Nutrients for nitrogen. Considering and addressing model uncertainties is crucial for interpreting and understanding the results accurately.

One approach to addressing model uncertainties is through comparative studies that assess the performance and consistency of various hydrological models. The ISIMIP community has published numerous papers focusing on model comparisons, including hydrological models. By referencing and discussing relevant studies from the ISIMIP community, the authors can contribute to the broader understanding of model uncertainties and provide insights into the robustness and reliability of their own model-based results.

Table 1

Could you please provide more context on how the thresholds are identified?

References

There are many format errors. Please correct.

To enhance the relevance and currency of the references in the present study, it would be beneficial to incorporate recent papers that demonstrate the research progress related to the topic. Regarding the old or unnecessary references, please remove or update them accordingly to ensure the accuracy and appropriateness of the reference list.

Reviewer #3 (Remarks to the Author):

The paper is a novel publication that estimates current and global water scarcity considering water quantity and quality. The article provides a comprehensive overview of the latest developments in the field, particularly with the novel future projections of water security.

Much of the existing work on water scarcity is focused on quantity. However, the degraded water quality poses barriers to water security and threatens human and ecosystem health and sustainable development. Thus, this paper is a positive contribution to the field and is worthy of publication with major revisions.

Another of the article's strengths is the integration of three different models into a modelling framework to determine regions under current and future water stress.

However, understanding the modelling framework and inputs required was challenging. My concern is the disparate methodology makes the work challenging to reproduce. I spend a significant amount of time digging through literature to comprehensively understand the model and methods. A more in-depth write-up in the supplemental of the models and the inputs would strengthen the article and improve reproducibility. More details are in the General comment section below.

Other than the issues I've highlighted below, I believe the way the presentation and discussion of the results require no further edits. I recommend accepting with major revisions.

General comments:

N Inputs and Mass Balance

1. MAGPIE is a land-system model, and N inputs and outputs are calculated based on model outputs. It considers diffuse sources such as synthetic fertilizers, animal manure and human excreta in agriculture, and leaching of organic matter from (non-)agricultural areas. Other papers go into the depths of the MAGPIE model methods. In this paper, the authors took the extra step of downscaling the modelled outputs. However, the authors gloss over these methodologies (especially the use of the LPJmL yield patterns), while in other papers, similar methods have been more extensively described (see Batool et al. (2022)). I encourage the authors to add more detail about these downscaling methods.

2. Inorganic fertilizer is disaggregated by considering crop demand and exogenous NUE. If so, please include a table of the regional NUE estimates.

The methods for the disaggregation of fertilizer are confusing. However, this approach to estimating inorganic fertilizer might be inaccurate in systems with large-scale industrial farming systems, which are also the areas most at risk of high N pollution export. Frequently, where crops are grown are not co-located where manure is produced ex. Spiegel et al. (2020) discuss this decoupling in the US, and Swaney et al. (2018) show how areas with large livestock operations impact NUE. In this case, you could be underestimating the applied N fertilizer. To verify these estimates, I suggest comparing them to other regional sources could help assess whether this approach is sufficient to estimate the disaggregation of inorganic fertilizer.

3. This article would benefit from N components (fertilizer, manure production, human waste) trajectories in the different scenarios (perhaps was to be included in the referenced "Table S6 in SI-II" but this table is not in the supplemental document.) Given that the N riverine loads are a function of N inputs and hydrology, I feel MARINA model inputs are not adequately presented. Specifically, I'm finding it challenging to understand how the inputs in each scenario lead to the different basin outcomes. I think maps or graphs with the N inputs and N removal (crop harvest) for all basins or aggregated by region (2010 and 2050) would help understand how different scenarios impact future water quality in different regions.

4. Lastly, it is unclear why China appears to have a specific manure discharge module (See Figure S2), whereas other countries do not. Some clarification on this would be helpful.

Model description

1. How is the MARINA model output in Eqn. 2 used to calculate the quality-based scarcity index? You reference the model output L in Eqn. 2, but in the supplemental, L is not in any equations.

2. There is minimal detail about the spin-up of the MARINA model and whether it considers the historical N use worldwide and, thus the existing legacy pools of N. How is N routed through the landscape? How is N surplus (downscaled MAGPIE inputs) used in MARINA? Is all remaining N after crop uptake assumed to be exported in that year's riverine load?

Legacy stores of N have been well documented in the literature (see Van Meter et al. (2017, 2018) and Ascott et al. (2017)). The paper would greatly improve with a caveat about how the results might change if landscape legacies are accounted for.

Manuscript Line Comments

Line 18: "Water scarcity is at stake today" should be "Water security is at stake today".

Line 31: "45.500 km³ year⁻¹" decimal point should be a comma.

Line 115: It should be S3 and S4, not S4 and S5.

Supplemental Line Comments

Line 70: Loads are estimated using "MARINA-Nutrients-Global-2.0" but version 2.0 is never mentioned. Is this a typo or a method that should be expanded upon?

Line 71: "Section 2.1.2" cited in the supplemental doesn't refer to any section in the paper.

Line 89: Equation 2 is missing from the document.

Line 160: Nitrogen use efficiency means different things to different disciplines. Please describe SNUPE and NUE.

Figure S2: Spelling mistakes in future.

Table S1: L source says "MARINA-Nutrients Global-1.0 (developed in this study, taking inputs from MAgPIE and VIC, a detailed description is available in the next section of this file)" but Figure 1 only has VIC used in Quantity-base scarcity estimates. Please clarify.

Figure S5 and S6: These figures are difficult to understand. The main text is opaque from where the numbers on line 108, "24%" and "76%," come from. I suggest reworking this figure for clarity.

References

Ascott, M. J., Goody, D. C., Wang, L., Stuart, M. E., Lewis, M. A., Ward, R. S., & Binley, A. M. (2017). Global patterns of nitrate storage in the vadose zone. In *Nature Communications* (Vol. 8, Issue 1). <https://doi.org/10.1038/s41467-017-01321-w>

Batool, M., Sarrazin, F. J., Attinger, S., Basu, N. B., Van Meter, K., & Kumar, R. (2022). Long-term annual soil nitrogen surplus across Europe (1850–2019). *Scientific Data*, 9(1), 1–22.

Van Meter, K. J., Basu, N. B., & Van Cappellen, P. (2017). Two centuries of nitrogen dynamics: Legacy sources and sinks in the Mississippi and Susquehanna River Basins. *Global Biogeochemical Cycles*, 31(1), 2–23.

Van Meter, K. J., Van Cappellen, P., & Basu, N. B. (2018). Legacy nitrogen may prevent the achievement of water quality goals in the Gulf of Mexico. *Science*, 360(6387), 427–430.

Response letter

Reviewer: 1

Comment 1:

The concept of the manuscript is very interesting and has scientific value. However, I have a few queries regarding some technical aspects.

Our response: We appreciate the reviewer for acknowledging the scientific value of our manuscript. We also thank the reviewer for the comments to improve our manuscript (see our responses to the comments below).

Comment 2:

The manuscript contains a significant number of acronyms. It would be helpful to introduce all the acronyms before the introduction section to improve the readability and understanding of the manuscript.

Our response: We agree with the reviewer that a list of acronyms will help the readers read the manuscript. In the original manuscript, we gave the full name for all acronyms for their first appearance in the text and in the caption each time they are used in a Figure/Table. In this revision, as suggested by the reviewer, we additionally provide a list of the acronyms as a separate file, which can be included before the Introduction or in the Supporting Information to be decided by the editorial office.

Comment 3:

The Abstract should be updated to reflect the tools and techniques utilized in the research, as well as highlighting the novelty of the study compared to other research in the field.

Line 21: Sentence should be revised in terms of language.

Our response: We thank the reviewer for this suggestion. We modified the Abstract to highlight our novelty and the integrated modelling approach that we used in this study. We also revised the language of the sentence in Line 21, as suggested by the reviewer. The updated text is as follows: "We conducted a **first global assessment** of future clean-water scarcity for 2050s by adding the water pollution aspect to the classical water quantity-induced scarcity assessments. This was done for >10,000 sub-basins focusing on nitrogen pollution in rivers by **integrating land-system, hydrological and water quality models.**" (see track changes in the Abstract in the revised manuscript).

Comment 4:

Introduction: The research objectives need to be clarified and stated more precisely. The authors can be presented them in bullet form.

Our response: We are sorry that the research objective was not clearly stated. The objective of this study is to assess future global "clean-water scarcity" in 2050 under climate and socio-economic changes. We modified the text to make this clearer (see track changes in the last paragraph of Introduction in the revised manuscript).

Comment 5:

Line 69: In this section, it would be beneficial for the authors to introduce the fundamental concept of the models used in their research and justify their choice for assessing water quality.

The authors should also address other water quality assessment models, such as the water quality index model, and compare them to the chosen model. To enhance the manuscript, I recommend adding a separate section on various water quality assessment models and rationalizing why the chosen model was used in the research by comparing it to other models.

Additionally, I have included a list of recent and relevant literature that could be helpful for improving the manuscript.

- 1) A review of water quality index models and their use for assessing surface water quality
- 2) A comprehensive method for improvement of water quality index (WQI) models for coastal water quality assessment
- 3) Assessing optimization techniques for improving water quality model
- 4) Assessment of urban river water quality using modified NSF water quality index model at Siliguri city, West Bengal, India
- 5) A novel approach for estimating and predicting uncertainty in water quality index model using machine learning approaches
- 6) A sophisticated model for rating water quality.
<https://doi.org/10.1016/j.scitotenv.2023.161614>

And As well as quantity-based model. Please see the recent documents

[1] Assessment of ecological water scarcity in China. DOI 10.1088/1748-9326/ac95b0

[2] Sensitivity of subregional distribution of socioeconomic conditions to the global assessment of water scarcity. DOI: <https://doi.org/10.1038/s43247-022-00475-w>

Our response: We agree with the reviewer that it is important to introduce the fundamental concept of the model and justify why MARINA-Nutrients was selected in this study for assessing quality-induced scarcity. We introduced the fundamental concept of the MARINA model in the Method section. To make this clearer, we modified the text in the last paragraph in Introduction (see track changes in the revised manuscript) to indicate model descriptions are available in Methods. We also added the justification of the models used in the Methods.

The reviewer provides a good suggestion to add a separate section to rationalize why MARINA-Nutrients was used in the research by comparing it to other models. We thus added a new paragraph in the Method section for this as follows:

“While developing this modelling framework, we also acknowledge that there are many other very useful models which could be adopted to assess clean-water scarcity. Many global hydrological models exist to assess water availability (Haddeland et al., 2011, van Vliet et al., 2016, Modi et al., 2022). The land use and agriculture model IMAGE, for example, is another well-known model in the field for simulating land use and nitrogen budget (Stehfest et al., 2014). Moreover, many water quality models exist to assess water pollution from various pollutants across temporal (e.g. daily, seasonal, annual) and spatial scales (e.g., catchment, sub-basin, basin). For example, SWAT (Arnold et al., 2012), WorldQual (Fink et al., 2018, Stehfest et al., 2014), and RTM (Liu et al., 2019) and IMAGE-GNM (Beusen et al., 2022) have been proven to be advanced models for assessing N pollution in

rivers. In this study, we decided to use the MARINA-Nutrients-Global-1.0 model as the starting point for clean-water assessment, in combination with MAgPIE and VIC, because of the following strengths. First, the MARINA model allows quantifying water pollution by source (i.e., fertilizer, manure, sewage and etc) which helps to better understand the main causes of water pollution. Second, the sub-basin approach of MARINA provides an opportunity to analyze water scarcity in large river basins in more detail. Moreover, not all models are available to assess future trends in clean-water scarcity under the SSPs and RCPs, while the combination of MAgPIE, VIC and MARINA is capable for such future assessment.” See track changes in the paragraph above Figure 1 of the revised manuscript.

We thank the reviewer very much for sharing several articles about the Water Quality Index (WQI) models, which are very interesting. We read the articles carefully. We find that the WQI models are different from the water quality model we use for our assessment. WQI serves as a powerful tool based on aggregation functions to convert varying water quality datasets to a single water quality index to assess the quality of the waterbody. This tool would need water quality data (e.g., nitrogen concentration in water) to be available. Our model, MARINA-Nutrients-Global-1.0, is a model that quantifies nitrogen load and concentration in rivers as the function of human activities in agriculture and wastewater treatment, and hydrology. The results of MARINA, together with other water quality data, can therefore be used as inputs to WQI to further calculate the overall water quality index for a water body. Although we cannot compare MARINA with WQI as they are different types of models, we see a great opportunity in applying the WQI approach to assess clean-water scarcity from the multi-pollutant perspective. Thus, we elaborated our discussion on the multi-pollutant water quality assessment by referring to the WQI models (see our response to Comment 12 by Reviewer 1).

The articles about quantity-based assessments suggested by the reviewer are also useful. Note that the assessment by Liu et al. (2022) was based on measured water availability and not based on models; we referred to this article where we show the overview of existing water scarcity assessment. We referred to Modi et al. (2022), where we justify the choice of our models (See track changes in the paragraph above Figure 1 of the revised manuscript).

Comment 6:

Line 73: The authors should report on the strengths and weaknesses of their model, including any uncertainties associated with the model outputs. This information is important for understanding the reliability and accuracy of the results.

Our response: We agree with this point. Reviewer 2 has provided the same suggestion. We thus added the strengths of the model in the text where we justify why MAgPIE, VIC and MARINA-Nutrients-Global-1.0 were selected to use in our study. We did this in the Method section due to limited space in the Introduction (see our response to Comment 5 by Reviewer 1), and because we would like to keep our Introduction focused on the knowledge gap “lack of global assessment of future clean-water scarcity”.

Regarding the weaknesses and uncertainties associated with the models, we added a separate section in the Discussion as follows:

“The integrated modelling approach. We assessed clean-water scarcity based on the integrated modelling framework (Figure 1) that links land use and agriculture, hydrological and water quality models. While the models provide great opportunities for exploring future trends and causes of water scarcity, there are uncertainties around the model inputs and modelling approaches of

MAGPIE, VIC and MARINA-Nutrients. Below we discuss why we consider our modelling approach reliable and sufficient in assessing future clean-water scarcity.

Uncertainties in our assessment from MAGPIE are mainly related to the estimated nitrogen (N) budgets (see Section “MAGPIE” in SI for the modelling approach). To build trust in our assessment, we compared our N budget from MAGPIE with a recent high-resolution (5 arc min degree) N dataset from Tian et al. (2022). The result shows a promising comparison. The global total nitrogen inputs to agriculture (cropland and pasture) and non-agriculture are very comparable, despite small differences among the sources as fertilizer, manure, and deposition (Table S8 in the SI). For example, the global total N input in 2010s is 267 Tg/year in Tian et al. (2022) and 287 Tg/year in this study. The spatial distribution of N inputs is also comparable between Tian et al. (2022) and our study (Figure S21 in SI). High total N inputs are observed in China, South Asia, Europe, United States and Brazil in both studies. There is an exception for atmospheric deposition. Some regions such as South Africa have higher deposition than in other regions in Tian et al. (2022) but not in our study. However, the N load quantified by the MARINA model is not sensitive to changes in atmospheric deposition due to its small contribution to water pollution compared to other sources. This was revealed in the thorough sensitivity analysis in Appendix E of Wang et al. (2022), $\pm 10\%$ changes in atmospheric deposition in the MARINA model hardly result in any difference in river export of N in MARINA. Considering the comparable total N inputs on land between the two studies, we believe that our results of quality-induced water scarcity hotspots will not change much when using N data from Tian et al. (2022). This comparison provides a high confidence in using the MAGPIE model for our assessment.

VIC’s simulated historical discharge and sectoral water withdrawals compared well against observed discharge (van Vliet et al., 2016) and reported domestic, industrial and irrigation water withdrawals (Droppers et al., 2019). However, multi-model intercomparison studies have shown that differences between hydrological models are the main source of uncertainty in discharge and irrigation water demand projections (Prudhomme et al., 2014, Schewe et al., 2014, Elliott et al., 2014). Therefore, the selection of a single model, the VIC hydrological model in our study impacts the water scarcity results. Nevertheless, we believe the model validation results show an acceptable performance of VIC for assessing water scarcity. VIC model estimates are mostly near the multi-model ensemble means for discharge (Haddeland et al., 2011) and irrigation (Wada et al., 2013). Moreover, the explicit representation of the energy balance in the VIC model allows the model to comprehensively capture the impacts of radiation changes under climate change, which is highly important for, for example, snow dynamics (Haddeland et al., 2011).

MARINA-Nutrients-Global-1.0 was developed and applied in this study to quantify N load at the sub-basin outlets. The previous versions of this model (Strokal et al., 2021, Wang et al., 2022, Wang et al., 2020) have been evaluated with a convincing performance at both global and regional scales based on the ‘building trust circle’ approach including 1) compare model outputs with measurements and existing studies, 2) compare spatial pattern of pollution hotspots, 3) sensitivity analysis, and 4) compare model inputs with independent datasets. The MARINA model, however, does not consider the legacy pools of N. Considering the historical N use can have a short-term impact on N pollution in groundwater (Ascott et al., 2017) and rivers (Van Meter et al., 2017), while the legacy effects balance out over longer time periods. If N legacy effects were considered, this would likely increase the modelled water pollution in most of the hotspot regions such as China, Europe, North America where historical N use was high, confirming our conclusion that water pollution will become an important cause of clean-water scarcity in these regions. Here, we consider that the MARINA version developed

in this study provides a robust assessment of quality driven-water scarcity because of the following reasons. First, the simulation of point source pollution from wastewater has been evaluated as promising in the global study of Strokal et al. (2021). Second, for diffuse source pollution we used data from MAgPIE, which shows good performance as discussed above when compared to N budget in other studies. Third, our modelled results compare very well with other global studies that quantify historical or future N pollution in rivers. For example, our global spatial patterns of N pollution are comparable to the total N patterns for 2000-2010 shown by van Vliet et al. (2021). This is the same for the future that we estimate similar hotspots of N pollution in 2010 and 2050 compared to Beusen et al. (2022).” (see track changes in the newly added Discussion-The integrated modelling approach in the revised manuscript).

Comment 7:

Line 196: “A similar situation is observed in North America”. Why?

Our response: The large industrial water withdrawal in North America is mainly due to the large water demand for energy production (e.g., thermoelectric power plant cooling) and manufacturing. We added this reason in the text to make this clearer (see track changes in the last sentence in Section 2.4 in the revised manuscript).

Comment 8:

Line 218 “Moreover, our assessment.....”. The statement should be revised in terms of language.

Our response: We revised the sentence to “Moreover, our assessment provides a more detailed view on non-hotspot areas by differentiating between low, moderate, and high levels of scarcity, considering both quantity and quality aspects.” (see track changes in the 2nd paragraph in Discussion in the revised manuscript)

Comment 9:

Line 316: “water scarcity indicators”. It would be helpful for readers if the authors could provide a few examples of the indicators used in the study, along with a justification for why they were selected.

Our response: We added in the text the examples of indicators and justification why one of them was selected as the basis to develop indicators for our study:

“Many water scarcity indicators consider water quantity, yet only a few consider both water quantity and water quality aspects (Liu et al., 2017, Liu et al., 2022, Liu et al., 2016, van Vliet et al., 2017, van Vliet et al., 2021, Ma et al., 2020). For example, the Quantity-Quality-EFR (QQE) indicator in Liu et al. (2016) was the first being developed and applied to assess both quality and quantity-based water scarcity based on the blue water footprint, grey water footprint, and environmental flow requirements (EFRs). Another example is the water quality dilution (WSq) indicator developed and applied globally to assess historical water scarcity as a proportion of sector-specific water withdrawals of suitable water quality to the total water availability (van Vliet et al., 2017, van Vliet et al., 2021, Ma et al., 2020). In this study we developed and applied clean-water scarcity indicators considering both aspects, inspired by the Quantity-Quality-EFR (QQE) indicator which has an advantage in quantifying the quality and quantity-based water scarcity separately. Therefore, these

newly developed indicators will enhance our understanding of overall water scarcity, aiding in the determination of whether the scarcity is predominantly due to issues of quantity or quality.”

See track changes in the 1st paragraph of Section “Methods” in the revised manuscript.

Comment 10:

Line 335: I have reviewed the modelling framework section and noticed that it may be difficult for readers to understand the attributes of the model, such as input variables, functions, and outputs in relation to assessing water quality. I recommend that the authors clearly define and present these attributes in a more precise and reader-friendly manner.

Our response: We are sorry that the modelling framework was not entirely clear to the reviewer. A detailed explanation of how each variable in Equation 1 and 2 for assessing clean-water scarcity was derived was given in Table S1 in the supporting information (SI). We also described the individual models in the modelling framework, including main equations, inputs for MARINA-Nutrients (interfaces between models: what outputs of MAgPIE and VIC are used in MARINA as inputs), description of MAgPIE and the downscaling approach in the SI (see Section “MARINA” and “MAgPIE” in the SI). Following comments by Reviewer 3 we have also made these descriptions in SI more complete and clear (see our responses to Comments 2-4 and 15 by Reviewer 3). We also edited the text in the Method Section to indicate clearly 1) which models are used in the modelling framework to assess the quantity-based and quality-based indicators, and 2) where to find detailed descriptions of the models (i.e., main equations, inputs, downscaling approaches) (see track changes in Section “Method”-“Modelling framework to assess clean-water scarcity”). We hope these revisions will help the readers to better understand the modelling approaches.

Comment 11:

Line 353: The MARINA-Nutrients model was used for water quality assessment in the research. However, it appears that this model only considers the nutrient components of water quality. It is worth noting that the overall quality of water depends on various indicators, such as dissolved oxygen, biological demand, pH, temperature, and heavy metal/trace metal/biological/hydrodynamics components. Therefore, it would be beneficial for the authors to provide some information and justification on why they chose to use this particular model and not other more reliable and advanced models that are currently available, such as the Irish Water Quality Index, National Sanitation Foundation, SRDD, etc. Adding such information would strengthen the overall quality of the research. I recommend some recent-relevant literature that could be helpful for improving this section.

- 1) A sophisticated model for rating water quality.
<https://doi.org/10.1016/j.scitotenv.2023.161614>
- 2) A review of water quality index models and their use for assessing surface water quality
- 3) Performance analysis of the water quality index model for predicting water state using machine learning techniques
- 4) A comprehensive method for improvement of water quality index (WQI) models for coastal water quality assessment
- 5) Development of river water quality indices—a review

Our response: We fully agree with the reviewer that the overall quality of water depends on various pollutants or indicators, not only limited to nitrogen pollution. We actually acknowledged in the

discussion in the original submission (Line 290 in the original manuscript): “We took nitrogen pollution as the water quality indicator in our study. However, many newly emerging pollutants (e.g., pathogens, antibiotics, plastics, and pesticides) will likely cause severe water degradation in the future (Strokal et al., 2021, Strokal et al., 2019, Li et al., 2022a, Strokal et al., 2022, Li et al., 2022b). Research is thus needed to identify the impacts of these pollutants on future water scarcity among sectors.”

The comments by the reviewer made us realize that we could have elaborated more here on what other approaches/ models could be useful (e.g., the water quality index suggested by the reviewer) to extend the assessment for other pollutants, temporal and spatial scales. We thus modified the text as follows:

“We took nitrogen pollution as the water quality indicator in our study. However, many other indicators (e.g., salinity, dissolved oxygen, biological demand, pH, temperature, and heavy metals) and newly emerging pollutants (e.g., pathogens, antibiotics, plastics, and pesticides) will likely cause severe water degradation in the future (Strokal et al., 2021, Strokal et al., 2019, Li et al., 2022a, Strokal et al., 2022, Li et al., 2022b). Research is thus needed to identify the impacts of these indicators or pollutants on future water scarcity among sectors as a joint effort of the water quality community. The advantage of our clean-water scarcity indicators is that they are not limited to specific pollutants and specific temporal or spatial scales. Therefore, the indicators can be quantified for various individual pollutants across temporal and spatial scales depending on the purpose of the assessment. Another opportunity is to combine our assessment approach with the Water Quality Index (WQI) models (Uddin et al., 2021, Parween et al., 2022, Uddin et al., 2023, Sutadian et al., 2016). WQI models are powerful tools based on aggregation functions to convert varying water quality datasets to a single water quality index to assess the quality of the waterbody. Such models can thus help to aggregate the quality-based indicators ($S_{quality}$) for individual pollutants to a simple single indicator for the water scarcity assessment.”

See track changes in the 3rd paragraph of Discussion- Implications for future water management and policies in the revised manuscript.

Comment 12:

Table 1: Is this classification scheme a newly proposed one or has it been proposed by someone else? If it came from other sources, could you please add the reference? If not, could you provide details on the methodology used to develop this scheme?

Our response: The classification scheme is newly proposed in this study based on water quality and water quantity thresholds from previous studies. We explained how the thresholds were chosen in the manuscript as follows:

“For quantity-based water scarcity ($S_{quantity}$, Equation 1), we used the generally accepted thresholds of >0.2 for moderate water scarcity and >0.4 for severe water scarcity. These levels were derived from existing studies (Liu et al., 2017, van Vliet et al., 2021, Alcamo and Henrichs, 2002, Hanasaki et al., 2018). For example, quantity-based water scarcity is considered severe when more than 40% of the available water – available river discharge after subtracting the amount for sustaining the environmental flow requirement - is used ($S_{quantity}>0.4$) (van Vliet et al., 2021). For quality-based water scarcity ($S_{quality}$, Equation 2), we used the thresholds of >0.45 for moderate water pollution and >1 for severe water pollution in view of avoiding eutrophication in aquatic ecosystems. The threshold of >0.45 means that TDN concentrations at the outlets of the sub-basins are higher than

0.45 mg N/L, indicating that the surface water bodies start to switch from the oligotrophic (clear water) to mesotrophic or eutrophic (turbid water) states, according to the trophic state index (TSI)(LCWA, 2022). The threshold of >1 means that nitrogen concentrations in the water bodies are higher than 1.0 mg N/L which can result in eutrophication in aquatic ecosystems (De Vries et al., 2013, Yu et al., 2019).”

We are sorry that we did not indicate in the caption of Table 1 where to find this information. We therefore modified the caption of Table 1 to indicate this clearly (see updated caption of Table 1 in the revised manuscript).

Comment 13:

Most of the labels for the tables and figures in the manuscript are too lengthy. It would be helpful to shorten them and make them more precise.

Our response: The reviewer is right that captions for many figures and tables are long. This is because we would like to provide adequate and complete information to make sure the figures and tables are understandable independently of the text to the readers, in line with the journals policy. We made the names of the scenarios shorter and clearer in the caption by indicating the SSPs-RCPs combinations (SSPs for socio-economic changes, RCPs for changes in hydrology induced by climate change). The details of the assumptions in the SSPs-RCPs scenarios are available in SI-II. We hope this helped.

Comment 14:

The authors should avoid the personal pronoun in manuscript.

Our response: We used the personal pronoun following the publishing style of Nature Communications. The use of personal pronouns “our study”, “we assessed” also help to highlight the contribution by us as authors of the manuscript. Nevertheless, we are happy to change this if the editor prefers us not to use personal pronouns.

Comment 15:

Reference should be updated using the recent-relevant literatures including suggested ones.

Our response: We updated the references and cited the most recent and suggested studies in the revised manuscript (see the updated reference list in the revised manuscript).

Reviewer: 2

Comment 1:

In the backdrop of the ever-evolving global landscape, water scarcity has emerged as a formidable impediment to the attainment of the Sustainable Development Goals (SDGs) set forth by the United Nations. Within the framework of SDG Target 6.4, it is explicitly stipulated that, "By 2030, there should be a substantial enhancement in water-use efficiency across all sectors, coupled with the establishment of sustainable practices for the extraction and provision of freshwater. These measures aim to address the pressing issue of water scarcity and significantly alleviate the plight of those afflicted by its repercussions." Although numerous studies have been dedicated to investigating water scarcity since the 1980s, the incorporation of water quality into water scarcity assessments remained a rarity until the advent of the 2010s. This research void was aptly recognized by the pioneering work of Junguo Liu and his associates, who spearheaded the integration of water quantity, water quality, and environmental flows in water scarcity assessments. Inspired by their groundbreaking contributions, several studies have emerged at various spatial scales, exemplified by the comprehensive global analysis by van Vliet et al. (2017) published in *Nature Geoscience* (10: 800–802), and the focused examination at the national level for China conducted by Ma et al. (2021) in *Nature Communications* (11: 650). Despite the commendable strides made thus far, a notable gap remains in terms of future water scarcity projections that effectively amalgamate quantity-based and quality-based assessments.

This paper unveils a captivating exploration into the realm of global clean water scarcity, encompassing a comprehensive analysis of water quantity, water quality, and environmental flows. Notably, this study presents a distinctive advancement by delivering a holistic assessment, encompassing both historical and prospective perspectives, of quantity- and quality-induced water scarcity across more than 10,000 sub-basins worldwide. A noteworthy revelation emerges from the research, emphasizing the imperative to shift our focus towards quality-induced water scarcity, as the findings reveal a threefold surge in basins grappling with future pollution-induced water scarcity on a global scale. This significant discovery underscores the urgency of devoting heightened attention to the intricate interplay between water quality and the alarming issue of water scarcity.

This study presents a compelling and captivating analysis, demonstrating meticulous attention to detail and a thoughtful examination of the subject matter. The manuscript exhibits a commendable structure that ensures readability and accessibility for readers. However, it is important to note that a few concerns warrant further consideration and should be diligently addressed in a revised version of the paper.

Our response: We thank the reviewer for the compliments that our manuscript presents a distinctive advancement and has significant discovery in the field of clean-water scarcity assessment. We are happy that the reviewer finds the analysis compelling and captivating, and the structure of the manuscript in commendable. Below we carefully addressed all comments by the reviewer.

Comment 2:

In lines 22-24, the comparison between different assessments, with and without consideration of quality-induced factors, lacks clarity. It only becomes apparent after reading

the entire manuscript. To improve clarity, please provide a more direct explanation of this comparison.

Our response: We agree with the reviewer and elaborated the Abstract highlighting that we compared the water scarcity with and without consideration of the quality-induced scarcity as follows (see track changes in the Abstract of the revised manuscript):

“We found that water pollution aggravates water scarcity in >2,000 sub-basins worldwide. The number of sub-basins with water scarcity triples due to future nitrogen pollution worldwide. In 2010, 984 sub-basins are classified as water scarce when considering only quantity-induced scarcity, while 2,517 sub-basins are affected by quantity & quality-induced scarcity. This number even increases to 3,061 sub-basins in the worst case scenario in 2050.”

Comment 3:

In lines 35-40, during the discussion on the impacts of climate change on the hydrological cycle and water resources, it is important to acknowledge a significant literature reference. The Intergovernmental Panel on Climate Change (IPCC) has released a report (WGII) specifically dedicated to addressing water-related issues. I strongly recommend that the authors thoroughly study this report and integrate relevant research findings and advancements into their discussion. This inclusion will enrich the analysis and provide a comprehensive overview of the current state of knowledge on this topic.

Caretta, M.A. et al., 2022. Water. In: *Climate Change 2022: Impacts, Adaptation, and Vulnerability. Contribution of Working Group II to the Sixth Assessment Report of the Intergovernmental Panel on Climate Change* [H.-O. Pörtner, D.C. Roberts, M. Tignor, E.S. Poloczanska, K. Mintenbeck, A. Alegría, M. Craig, S. Langsdorf, S. Löschke, V. Möller, A. Okem, B. Rama (eds.)]. Cambridge University Press, Cambridge, UK and New York, NY, USA, pp. 551-712, doi:10.1017/9781009325844.006.

Our response: We thank the reviewer for sharing that there are important findings by the latest IPCC report. We find the main conclusions of the report on water scarcity are in line with our main conclusion that close to half of the global population currently live in quantity-induced water scarcity regions. The message from the report that water quality will become an important cause of water scarcity is also consistent with ours. In our discussion, we actually already cited several important references in the report including (Gosling and Arnell, 2016, Mekonnen and Hoekstra, 2016, van Vliet et al., 2021, Liu et al., 2017). We added the above report (Caretta et al., 2022) suggested by the reviewer and a few more important references (Koutroulis et al., 2019, Modi et al., 2022, Greve et al., 2018, Seneviratne et al., 2021) from the report in our elaborated Discussion and Introduction (see updated reference list and the revised Discussion and Introduction in the revised manuscript).

Comment 4:

Line 50-52: It is recommended to provide references to support these statements to reinforce the validity of the claims made.

Our response: We added references (listed below) that support the statement that so far water scarcity assessments have focused mainly on changes in water quantity (see added references in the 1st sentence of the 3rd paragraph of the revised manuscript).

- Gosling, S. N. & Arnell, N. W. A global assessment of the impact of climate change on water scarcity. *Climatic Change* 134, 371-385, doi:10.1007/s10584-013-0853-x (2016).

- Liu, J., Yang, H., Gosling, S. N., Kummu, M., Flörke, M., Pfister, S., Hanasaki, N., Wada, Y., Zhang, X. & Zheng, C. Water scarcity assessments in the past, present, and future. *Earth's future* 5, 545-559 (2017).
- Schewe, J., Heinke, J., Gerten, D., Haddeland, I., Arnell, N. W., Clark, D. B., Dankers, R., Eisner, S., Fekete, B. M., Colón-González, F. J., Gosling, S. N., Kim, H., Liu, X., Masaki, Y., Portmann, F. T., Satoh, Y., Stacke, T., Tang, Q., Wada, Y., Wisser, D., Albrecht, T., Frieler, K., Piontek, F., Warszawski, L. & Kabat, P. Multimodel assessment of water scarcity under climate change. *Proceedings of the National Academy of Sciences* 111, 3245-3250, doi:doi:10.1073/pnas.1222460110 (2014).
- Cui, R. Y., Calvin, K., Clarke, L., Hejazi, M., Kim, S., Kyle, P., Patel, P., Turner, S. & Wise, M. Regional responses to future, demand-driven water scarcity. *Environmental Research Letters* 13, 094006 (2018).
- Greve, P., Kahil, T., Mochizuki, J., Schinko, T., Satoh, Y., Burek, P., Fischer, G., Tramberend, S., Burtscher, R., Langan, S. & Wada, Y. Global assessment of water challenges under uncertainty in water scarcity projections. *Nature Sustainability* 1, 486-494, doi:10.1038/s41893-018-0134-9 (2018).
- Koutroulis, A. G., Papadimitriou, L. V., Grillakis, M. G., Tsanis, I. K., Warren, R. & Betts, R. A. Global water availability under high-end climate change: A vulnerability based assessment. *Global and Planetary Change* 175, 52-63, doi:https://doi.org/10.1016/j.gloplacha.2019.01.013 (2019).

Comment 5:

Line 52-57: This section lacks key references, including an early study published in 2010, which investigated spatial patterns of 11 nitrogen inputs/outputs on a global scale with high spatial resolution. Additionally, a recent publication by Tian et al. (2022) on high-resolution nitrogen inputs is also missing. It is crucial to include these references to ensure a comprehensive analysis of the topic.

Moreover, it would be beneficial to compare the results obtained in the present study with those of Tian's study. This comparison will allow the authors to highlight the advantages of the nitrogen method employed in their research in contrast to Tian's approach.

- Liu J., You L.Z., Amini M., Obersteiner M., Herrero M., Zehnder A.J.B., Yang H. 2010. A high-resolution assessment on global nitrogen flows in cropland. *Proceedings of the National Academy of Sciences of the United States of America* 107(17): 8035-8040.
- Tian H., et al., 2022. History of anthropogenic Nitrogen inputs (HaNi) to the terrestrial biosphere: a 5-arcmin resolution annual dataset from 1860 to 2019. *Earth System Science Data* 14 (10): 4551-4568.

Our response: We thank the reviewer for providing the above insightful studies. As suggested by the reviewer, we cited the suggested articles in the manuscript (see added references in the 3rd paragraph in the revised manuscript).

We also compared the data of nitrogen inputs from Tian et al. (2022) with the N inputs in this study based on MAgPIE (Model of Agricultural Production and its Impacts on the Environment) for 1990s, 2000s and 2010s. The results are summarized in Table S8 and Figure S21 in the Supporting Information (SI) and show a promising comparison as discussed below:

“The global total nitrogen inputs to agriculture (cropland and pasture) and non-agriculture are very comparable, despite small differences among the sources as fertilizer, manure, and deposition (Table S8 in the SI). For example, the global total N input in 2010s is 267 Tg/year in Tian et al. (2022) and 287 Tg/year in this study. The spatial distribution of N inputs is also comparable between Tian et al. (2022) and our study (Figure S21 in SI). High total N inputs are observed in China, South Asia, Europe, United States and Brazil in both studies. There is an exception for atmospheric deposition. Some regions such as South Africa have higher deposition than in other regions in Tian et al. (2022) but not in our study. However, the N load quantified by the MARINA model is not sensitive to changes in atmospheric deposition due to its small contribution to water pollution compared to other sources.

This was revealed in the thorough sensitivity analysis in Appendix E of Wang et al. (2022), $\pm 10\%$ changes in atmospheric deposition in the MARINA model hardly result in any difference in river export of N in MARINA. Considering the comparable total N inputs on land between the two studies, we believe that our results of quality-induced water scarcity hotspots will not change much when using N data from Tian et al. (2022). This comparison provides a high confidence in using the MAGPIE model for our assessment.”

We added the above discussion to the Discussion Section for building trust of our modelling approach (see track changes in the 2nd paragraph Section Discussion-“The integrated modelling approach” in the revised manuscript).

Comment 6:

Line 58-63: There is an opportunity to enhance the presentation of research progress in this section. While references 21-22 focus on regional studies, it is worth noting that they are pioneering work on water scarcity assessment with consideration for water quality. By conducting a thorough investigation of subsequent studies, it becomes evident that most global-scale studies align with the fundamental principles outlined in references 21-22, with similar methodologies being employed.

By highlighting these aspects, the authors can emphasize the foundational contributions of references 21-22 and demonstrate the consistency in approaches taken by subsequent studies on a global scale.

Our response: We agree with the reviewer. It is very important to show the research progress in the field of water scarcity assessment. We modified the text as suggested by the reviewer to “Two pioneering studies have quantified regional water scarcity by innovatively integrating assessments of both water quality and quantity (Liu et al., 2016, Zeng et al., 2013). Van Vliet (van Vliet et al., 2017, van Vliet et al., 2021) have been the first to assess water scarcity on a global scale using a sector-specific approach focusing on multiple pollutants including nitrogen for the historical period of 2000-2010.” (see track changes in the 4th paragraph of the revised manuscript)

Comment 7:

In line 88, it is unnecessary to repeat the definition of water scarcity, as it has already been provided earlier in the manuscript.

Our response: As suggested by the reviewer, we deleted the definition in the text to avoid repetition (see track changes in the 1st paragraph of Section 2.1 in the revised manuscript).

Comment 8:

Lines 159-171: While the paper emphasizes the global changing trends in water scarcity, it would also be valuable to explore the specific trends in different countries, such as China, India, and the US. Understanding the variations and patterns of water scarcity in these countries can provide further insights into regional dynamics and help tailor localized strategies and interventions. Therefore, it is recommended to consider including a discussion on the trends and implications of water scarcity in specific countries of interest, along with their unique challenges and potential solutions.

Our response: We thank the reviewer for this nice suggestion. We agree that it is important to emphasize the regional trends in water scarcity as the main causes/drivers differ among regions. We

did this for China, India and other regions as the continental scale because of their similarities in the future trend. Considering the comment of the reviewer we have elaborated more and discussed clearly the unique challenges of addressing clean-water scarcity among regions. We have added an additional section in Results to discuss the different challenges among hotspots.

“2.4 Different challenges among hotspots

While future hotspots of clean-water scarcity are identified mainly in China, India, Europe, North America and in the worst-case scenario (SSP5-RCP8p5) also in Central Africa. The causes of water scarcity differ among these regions, presenting different challenges that need to be addressed to reduce water scarcity.

For quantity-induced scarcity, the main causes are the excessive withdrawals (high water withdrawal over water availability). The share of water withdrawals among sectors varies largely across continents (Figure S12 in SI). Irrigation contributes most of surface water withdrawals on the global scale and is the most important driver of quantity-induced scarcity in most regions including China, India and South America. However, this differs in Europe, where irrigation contributes to less than 30% of water withdrawals. The most important water withdrawal is the industrial sector in Europe. A similar situation is observed in North America, where the industry takes almost 50% of the continental water withdrawal due to the large water demand for energy production (e.g., thermoelectric power plant) and manufacturing (Droppers et al., 2019).

For quality-induced scarcity, the main causes of high TDN inputs to rivers are also different among hotspots. In 2010, TDN inputs to rivers are mainly driven by the low nitrogen use efficiencies in China and India (Table S3 in SI), high production in Europe and North America (Figure S19 in SI), and by atmospheric N deposition and fixation on natural land in South America and Central Africa. In the future, the main cause of TDN inputs to rivers is similar across most hotspots in SSP1 and SSP2, which is agricultural production. It is important to note that although the nitrogen use efficiencies have improved to high levels, the high food production in China, India, Europe and North America (e.g., Mississippi river) driven by food demand still leads to high N surpluses in agriculture (Figure S20 in SI). In SSP5, pollution is driven worldwide mainly by sewage as described above due to global urbanization and inadequate development of sewage treatment. Atmospheric N deposition and fixation on natural land remain the main source of TDN rivers in South America and Central Africa in the future, while agricultural N surpluses become increasingly more important in SSP5 (Figure S20 in SI).” (see track changes in Section 2.4 in the revised manuscript)

Comment 9:

Line 185: 4355%? How come?

Our response: This big difference is due to increase in river discharge in very dry regions. We realize this expression is confusing so we modified the text to “The changes among the individual sub-basins vary from -156 to +117 km³/year between 2010 and 2050” (see track changes in the 2nd paragraph in Section 2.3 in the revised manuscript)

Comment 10:

Line 205-205: The statements made in this section are not entirely accurate. “Clean water scarcity” is not a new concept but rather a recent terminology used to emphasize the importance of addressing both the quantity and quality aspects of water scarcity.

Our response: We agree with the reviewer that “clean-water scarcity” is not a new concept but rather a new terminology. In previous studies, there was no terminology for this concept, and rather referred it as water scarcity considering both water quality and quantity. We think it is nice to introduce this terminology with a clear definition so that it can be easily referred to in future studies. We modified the text and referred “clean-water scarcity” as a new terminology based on the concept developed by previous studies (see track changes in the 1st paragraph of Section Discussion in the revised manuscript).

Comment 11:

Line 261-262 and other related sections on environmental flow requirements: It is crucial to recognize that different river basins have varying levels of environmental flow requirements. Several studies have been conducted on this topic, providing valuable insights into the specific needs of individual river basins. To enhance the accuracy and effectiveness of the assessment, it is advisable to incorporate state-of-the-art findings on environmental flow requirements. By considering these findings, the study can avoid applying a single value for all situations, such as the mentioned 37%, and instead adopt a more nuanced approach that accounts for the unique characteristics and ecological demands of each river basin.

Our response: We fully agree that the reviewer has a good point that environmental flow requirements (EFRs) vary among the river basins, as we acknowledged in our Discussion section of the submitted manuscript as well. We took this opportunity and calculate the annual EFRs following the approach of Pastor et al. (2014) as described in the SI: “We derived the annual EFRs for sub-basins following the approach of Pastor et al. (2014) using mean annual flow (MAF) and mean monthly flow (MMF) provided by the VIC (Variable Infiltration Capacity) model runs (van Vliet et al., 2016). We first determined the hydrological seasons as follows: low-flow month when $MMF \leq 0.4 \cdot MAF$, high-flow month when $MMF > 0.8 \cdot MAF$, intermediate-flow month when $0.4 \cdot MAF < MMF \leq 0.8 \cdot MAF$. Next, we calculated the EFRs based on the hydrological seasons: EFRs for low-flow months are estimated as $0.6 \cdot MMF$, for high-flow months are $0.3 \cdot MMF$, for intermediate-flow months are $0.45 \cdot MMF$. Last, we aggregate the monthly EFRs (from last step) to annual EFRs as a fraction of MAF. We derived the EFRs for 2010 and 2050. Results vary between 30% and 38% among sub-basins (see Figure S9). This is comparable to the conclusion of Pastor et al. (2014) who suggested reserving an annual average of 37% of global annual flows to keep ecosystems in a fair ecological condition for global water availability assessments.”

The resulted annual EFRs vary between 30 and 38% among the sub-basins (Figure S9 in SI). We updated the clean-water scarcity assessment with these sub-basin specific EFRs. The results, however, do not show big differences in the number of clean-water scarcity hotspots (e.g., 2523 vs 2517 before and after updating EFRs in 2010) (Figure 2). The main reason for this is that we performed this assessment at an annual scale at which EFRs do not vary much. Our clean-water scarcity indicators could also be applied to other temporal scales such as monthly scale if data is available. In such a case we believe that the larger difference in sub-basin-specific EFRs may result in a different clean-water scarcity assessment.

We updated all figures and tables, and numbers in the text of the manuscript after this recalculation (see all new figures and tables, and track changes in the revised manuscript).

Comment 12:

Line 284-285: When discussing food waste and losses, it is important to provide references to support the statements made. Over the past years, there have been significant research efforts dedicated to this topic, resulting in a wealth of literature on the subject. By incorporating relevant references, the paper can strengthen the credibility of its discussion on food waste and losses and ensure that the information presented is backed by established research in the field.

Our response: We thank the reviewer for this suggestion and cited several important recent studies about N management from the perspectives of food waste, food demand and production (Grizzetti et al., 2013, Springmann et al., 2018, Kanter et al., 2020a, Houlton et al., 2019) in the manuscript (see updated references in the 2nd paragraph of Section “Implications for future water management and policies” in the revised manuscript).

Comment 13:

It would be beneficial for the authors to discuss the uncertainties associated with the models utilized in their study, namely VIC for water and MARINA-Nutrients for nitrogen. Considering and addressing model uncertainties is crucial for interpreting and understanding the results accurately. One approach to addressing model uncertainties is through comparative studies that assess the performance and consistency of various hydrological models. The ISIMIP community has published numerous papers focusing on model comparisons, including hydrological models. By referencing and discussing relevant studies from the ISIMIP community, the authors can contribute to the broader understanding of model uncertainties and provide insights into the robustness and reliability of their own model-based results

Our response: Reviewer 1 had a similar comment. We agree with the reviewer that it is important to discuss the uncertainties associated with the models and their use in this study. We therefore added an extra sub-section in Discussion to discuss the model performances of VIC, MARINA and MAgPIE. As suggested by the reviewer, we referenced and discussed studies from the ISIMIP community. We believe that the newly added discussion contributes to building trust of the robustness of our modelling assessment.

For details, see our response to Comment 6 of Reviewer 1.

Comment 14:

Table 1

Could you please provide more context on how the thresholds are identified?

Our response: Reviewer 1 had the same question. See our response to Reviewer 1, Comment 12.

Comment 15:

References

There are many format errors. Please correct.

To enhance the relevance and currency of the references in the present study, it would be beneficial to incorporate recent papers that demonstrate the research progress related to the topic. Regarding the old or unnecessary references, please remove or update them accordingly to ensure the accuracy and appropriateness of the reference list.

Our response: We corrected the references and cited the most recent studies in the revised manuscript (see the updated reference list in the revised manuscript).

Reviewer 3

Comment 1:

The paper is a novel publication that estimates current and global water scarcity considering water quantity and quality. The article provides a comprehensive overview of the latest developments in the field, particularly with the novel future projections of water security.

Much of the existing work on water scarcity is focused on quantity. However, the degraded water quality poses barriers to water security and threatens human and ecosystem health and sustainable development. Thus, this paper is a positive contribution to the field and is worthy of publication with major revisions.

Another of the article's strengths is the integration of three different models into a modelling framework to determine regions under current and future water stress.

However, understanding the modelling framework and inputs required was challenging. My concern is the disparate methodology makes the work challenging to reproduce. I spend a significant amount of time digging through literature to comprehensively understand the model and methods. A more in-depth write-up in the supplemental of the models and the inputs would strengthen the article and improve reproducibility. More details are in the General comment section below.

Other than the issues I've highlighted below, I believe the way the presentation and discussion of the results require no further edits. I recommend accepting with major revisions.

Our response: We thank the reviewer for the compliments that our manuscript is novel and provides a comprehensive overview of the latest developments in the field. We are happy that the reviewer applauds the integrated modelling framework that we developed to assess clean-water scarcity. We carefully addressed all comments by the reviewer. We appreciate that the reviewer indicated that the way the presentation and discussion of the results require no further edits. As suggested by the reviewer, we elaborated the model description in the supporting information to a more in-depth description. The point-by-point responses are available below.

Comment 2:

N inputs and Mass Balance: MAGPIE is a land-system model, and N inputs and outputs are calculated based on model outputs. It considers diffuse sources such as synthetic fertilizers, animal manure and human excreta in agriculture, and leaching of organic matter from (non-)agricultural areas. Other papers go into the depths of the MAGPIE model methods. In this paper, the authors took the extra step of downscaling the modelled outputs. However, the authors gloss over these methodologies (especially the use of the LPJmL yield patterns), while in other papers, similar methods have been more extensively described (see Batool et al. (2022)). I encourage the authors to add more detail about these downscaling methods.

Our response: We really appreciate that the reviewer pointing this out. We realize that we should describe more clearly how we downscaled the MAGPIE results to 0.5° grid level. We have now added a detailed description in the Supporting Information (see track changes in the 6th-9th paragraphs in the Section "MAGPIE" in the revised SI) that describes the downscaling as follows:

"Land patterns are downscaled from cluster to 0.5° as follows: First, for all land types that show a reduction of area from one time step to another, we calculate a relative reduction factor for each low-

resolution cluster. Second, we apply the cluster-level reduction factor to the 0.5° land patterns of the previous timestep (using historical data for the first projection timestep). Third, we calculate for all expanding land pools the share of each individual expanding land type in the sum of all expanding land types in a cluster cell. Fourth, we use this share to fill the area that was reduced in step 2. This method makes sure that the total of each land pool is identical on 0.5° and on cluster level, while no land pool exceeds the cell size (which could happen if proportional changes were applied to the previous 0.5° pattern).

After crop area was downscaled to 0.5°, we downscaled production under consideration of the relative yield patterns at 0.5° within each cluster. The relative yield patterns are derived using the crop yield simulations from the LPJmL model that assume homogeneous management intensity within all 0.5° grid cells. We multiply the downscaled crop area with the yields under homogenous management to derive a production potential. We use this potential as disaggregation weight to derive 0.5° production patterns from cluster-level production projections. These differ in sum from the production potential under homogenous management as current production stays sometimes below the production potentials, but also as the MAgPIE model simulates future yield improvements that can extend current production potentials.

Manure availability is disaggregated differently for ruminant and monogastric livestock. Manure of ruminants is split based on the nitrogen feed intake into a pasture and a cropland fraction. The pasture fraction is assumed to be excreted on pastures based on pasture production, such that higher-yielding pastures also receive more manure. The cropland fraction was disaggregated using Nr in crop production as weight (yet excluding second generation bioenergy crops which are not used as feed). Manure allocation for monogastric livestock differs by economic development. In low-income regions, we disaggregate the manure to 0.5° using the built-up area as disaggregation weight because in low-income countries monogastrics are mostly kept in extensive systems close to the human population (Gilbert et al., 2015). In high-income regions, we assumed that monogastric production takes place close to the feed production, so we used cropland as disaggregation weight. For middle-income regions, we interpolate between both approaches.

After downscaling crop area, crop production and manure availability, we recalculate the harvested nitrogen and organic inputs on 0.5° resolution using the method described in (Bodirsky et al., 2014, Bodirsky et al., 2012). For example, Nr in crop harvest is estimated using downscaled production and crop-specific nitrogen contents, and biological fixation by free-living nitrogen fixers is estimated based on downscaled crop area.

Because no (non-modeled) global subnational data on fertilizer application exists, we assume that farmers apply fertilizer proportional to plant requirements under consideration of available organic fertilizers. We first calculate the soil uptake requirements by the plants. As the availability of organic fertilizers may be unequally distributed with availability exceeding plant requirements at places with high manure and crop residue availability, we top-up the organic by inorganic fertilizer primarily at those places where the SNU_pE (croplands) or NUE (pastures) would be highest without the addition of fertilizers. This fertilizer distribution is done such that we minimize the maximum SNU_pE across 0.5° grid cells, while allowing for a low SNU_pE in places with high organic fertilizer availability. Redistribution of organic fertilizers, e.g. by the transport of manure, is not simulated.”

Comment 3:

N inputs and Mass Balance: Inorganic fertilizer is disaggregated by considering crop demand and exogenous NUE. If so, please include a table of the regional NUE estimates.

Our response: Inorganic fertilizer is estimated as the necessary budget closure to fulfill plant requirements under a given SNU_pE (soil nitrogen uptake efficiency) for croplands or NUE (nitrogen use efficiency) for pastures under given quantities of organic fertilizers. As suggested by the reviewer, we added a table with exogenous assumptions on regional SNU_pE for each scenario and regional NUE that is the same for all scenarios in Table S3 in the revised SI.

See track changes in the 3rd paragraph of Section “MAGPIE” in the revised SI for definitions of SNU_pE and NUE, and the approach for estimating organic fertilizers.

Comment 4:

N inputs and Mass Balance: The methods for the disaggregation of fertilizer are confusing. However, this approach to estimating inorganic fertilizer might be inaccurate in systems with large-scale industrial farming systems, which are also the areas most at risk of high N pollution export. Frequently, where crops are grown are not co-located where manure is produced ex. Spiegel et al. (2020) discuss this decoupling in the US, and Swaney et al. (2018) show how areas with large livestock operations impact NUE. In this case, you could be underestimating the applied N fertilizer. To verify these estimates, I suggest comparing them to other regional sources could help assess whether this approach is sufficient to estimate the disaggregation of inorganic fertilizer.

Our response: To make more clear how inorganic fertilizers were downscaled, we added the following paragraph (see track changes in the last two paragraphs in the Section “MAGPIE” in the revised SI):

“Because no (non-modeled) global subnational data on fertilizer application exists, we assume that farmers apply fertilizer proportional to plant requirements under consideration of available organic fertilizers. We first calculate the soil uptake requirements by the plants. As the availability of organic fertilizers may be unequally distributed with availability exceeding plant requirements at places with high manure and crop residue availability, we top-up the organic by inorganic fertilizer primarily at those places where the SNU_pE (croplands) or NUE (pastures) would be highest without the addition of fertilizers. This fertilizer distribution is done such that we minimize the maximum SNU_pE across 0.5° grid cells, while allowing for a low SNU_pE in places with high organic fertilizer availability. Redistribution of organic fertilizers, e.g. by the transport of manure, is not simulated.

Cropping systems with leguminous crops often have a higher NUE (Lassaletta et al., 2014, Smil, 1999, Swaney et al., 2018), as biological fixation occurs within plant roots and is not subject to leaching, denitrification and volatilization before harvest. Our disaggregation approach for croplands takes account of this by using SNU_pE ((Harvest-Biological Fixation-Seed) / (Organic and Inorganic Soil Inputs)) instead of NUE (Harvest/(Organic and Inorganic Soil Inputs + Biological Fixation)). With the same SNU_pE, 0.5° grid cells with leguminous crops therefore have a higher NUE and receive less inorganic fertilizer than cells with without symbiotic nitrogen fixation. For pastures, we use NUE as we cannot separate symbiotic fixation within plants from the fixation by free-living organisms. Cropping systems with high manure inputs have lower requirements of inorganic fertilizer due to substitution, yet also a lower NUE (Swaney et al., 2018). Our method accounts for the substitution effect, but not for the lower fertilizer equivalence of manure. This, in combination with the rather

homogenous distribution of livestock across croplands, leads to an underestimation of the heterogeneity of the Nr losses in space.”

Comment 5:

This article would benefit from N components (fertilizer, manure production, human waste) trajectories in the different scenarios (perhaps was to be included in the referenced "Table S6 in SI-II" but this table is not in the supplemental document.) Given that the N riverine loads are a function of N inputs and hydrology, I feel MARINA model inputs are not adequately presented. Specifically, I'm finding it challenging to understand how the inputs in each scenario lead to the different basin outcomes. I think maps or graphs with the N inputs and N removal (crop harvest) for all basins or aggregated by region (2010 and 2050) would help understand how different scenarios impact future water quality in different regions.

Our response: The reviewer has a very good point that information on the assumptions in N inputs among the scenarios will help to better understand how different scenarios impact future water quality in different regions. We summarized the assumptions for the scenarios with references in Tables in SI-II. We are sorry that this file was missing in the first round submission. We added file SI-II with the table in this submission (see Tables S9-11 in SI-II).

As suggested by the reviewer, we also added maps of the N inputs from fertilizer, N inputs from manure, total N inputs to agriculture, N uptake by crops, N balance in agriculture (total input – N uptake) for all basins in 2010 and 2050 in the Supplementary Information (see new Figures S16-S20 in the revised SI). Note that changes in water pollution in the future are also driven by human waste and its treatment, we thus also added new maps for N inputs to rivers from sewage systems and open defecation in 2010 and 2050 for all sub-basins (see new Figures S13-S15 in the revised SI) The original data for making these maps will also be uploaded to an open-access online repository upon acceptance of the manuscript.

Comment 6:

Lastly, it is unclear why China appears to have a specific manure discharge module (See Figure S2), whereas other countries do not. Some clarification on this would be helpful.

Our response: The reviewer raised a very good question. We actually explained this in Table S2 of the Supporting Information: “Studies show that in 2010, China had poor manure management (Strokal et al., 2016, Wang et al., 2018, Wang et al., 2020). Part of the collected animal manure were not treated properly or reused in cropland but were discharged to rivers directly. Thus, for the Chinese sub-basins in 2010, we took data for the direct discharge of animal manure from the MARINA-Nutrients-China-2.0. For 2050, we assume that there is no direct discharge of animal manure in China according to the recently introduced manure management regulations by Chinese government. Thus for 2050, we used the data on manure management from model input category 1.” To make it clearer, we modified the caption of Figure S2 the explanation for the specific manure discharge in China can be found in Table S2 (see track changes in update caption of Figure S2).

Comment 7:

How is the MARINA model output in Eqn. 2 used to calculate the quality-based scarcity index? You reference the model output L in Eqn. 2, but in the supplemental, L is not in any equations.

Our response: These questions by the reviewer made us realize that it was not clear enough in the original Methods and Supporting Information which outputs of MARINA were used as pollutant load (L) in Equation 2. The water quality-based indicator ($S_{quality}$ in Equation 2 in the main text) can be used to assess quality-driven water scarcity resulted from any pollutants (L as pollutant load in Equation 2) in water systems. In this study, L is based on the total dissolved nitrogen (TDN) load at sub-basins outlets simulated by MARINA-Nutrients-Global-1.0. The equations to quantify TDN load in MARINA are summarized in Equations S3 and S4 (referred as $OT_{F,y,j}$ in Equation S3 for individual rivers or tributaries, and $OC_{F,y,j}$ in Equation S4 for main channel). See Figure S1 for the definition of tributary and main channel in MARINA. We modified the text in Methods, and in the SI to make this clearer to the readers (see track changes in the paragraph below Equation 2, in the 3rd paragraph of the Section “Method – Modelling framework to assess clean-water scarcity” in the revised manuscript; see track changes in the Sections “Data for clean-water scarcity” and “MARINA-Nutrients-Global-1.0” in the revised SI).

Comment 8:

There is minimal detail about the spin-up of the MARINA model and whether it considers the historical N use worldwide and, thus the existing legacy pools of N. How is N routed through the landscape? How is N surplus (downscaled MAGPIE inputs) used in MARINA? Is all remaining N after crop uptake assumed to be exported in that year's riverine load?

Legacy stores of N have been well documented in the literature (see Van Meter et al. (2017, 2018) and Ascott et al. (2017)). The paper would greatly improve with a caveat about how the results might change if landscape legacies are accounted for.

Our response: The reviewer raised very good questions about the MARINA model. Regarding the use of N surplus from MAGPIE: MARINA quantifies N inputs to rivers from agricultural sources based on N inputs, uptake, and retention on land as a function of runoff (see text below Equation S2 in “MARINA-Nutrients Global 1.0” in SI).

The MARINA model does not take into account the landscape legacies. Although historical nitrogen use can have a short-term impact on N pollution in groundwater (Ascott et al., 2017) and rivers (Van Meter et al., 2017), we believe this will not influence the main conclusion of this study. This is mainly because the main purpose of this study is to explore changes in water pollution in a relatively long-term future in 2050 compared to 2010 as impacts of future socio-economic and climate changes. And even if we consider the N legacy, this will probably increase the modelled water pollution in most of the hotspot regions such as China, Europe, North-America where historical N use was high, confirming our conclusion that water pollution will become an important cause of clean-water scarcity. We added the above discussion in our Discussion section, where we discuss the uncertainties of the MARINA approach (see track changes in Discussion-The Integrated Modelling Approach in the revised manuscript).

Comment 9:

Line 18: "Water scarcity is at stake today" should be "Water security is at stake today".

Our response: We agree with the reviewer and changed the text accordingly (see track changes in the Abstract of the revised manuscript).

Comment 10:

Line 31: "45.500 km³ year⁻¹" decimal point should be a comma.

Our response: We thank the reviewer for pointing out this mistake and corrected this in the manuscript (see track changes in the 1st paragraph in the Introduction section of the revised manuscript).

Comment 11:

Line 115: It should be S3 and S4, not S4 and S5.

Our response: We thank the reviewer for pointing out this mistake. We meant to refer to Figures S5 and S6. We corrected this in the manuscript and now refer to the correct Figures (see track changes in the 3rd paragraph in Section 2.1 of the revised manuscript).

Comment 12:

Supporting information: Line 70: Loads are estimated using "MARINA-Nutrients-Global-2.0" but version 2.0 is never mentioned. Is this a typo or a method that should be expanded upon?

Our response: We thank the reviewer for pointing out this typo. This should be MARINA-Nutrients-Global-1.0, as mentioned in other parts of the manuscript. We corrected this in the Supporting Information (SI) (see track changes in the section 'Data for clean-water scarcity assessment' of the revised SI).

Comment 13:

Supporting information: Line 71: "Section 2.1.2" cited in the supplemental doesn't refer to any section in the paper.

Our response: We are sorry for this confusion. We meant to refer to the Section "MARINA-Nutrients-Global-1.0" in the Supporting Information (SI). In this section, we describe the main equations of the MARINA-Nutrients model. Equations S3 and S4 were used to calculate the actual N load at the outlet of sub-basins (Equation S3 for individual rivers, Equation S4 for main channels). We edited the text in the text of SI to make this clearer (see track changes in the section "MARINA-Nutrients Global 1.0" of the revised SI).

Comment 14:

Supporting information: Line 89: Equation 2 is missing from the document.

Our response: We are sorry for this confusion. We corrected the numbering of the equation in the Supporting Information (see track changes in Equations S1-S4 of the revised SI).

Comment 15:

Supporting information: Line 160: Nitrogen use efficiency means different things to different disciplines. Please describe SNUPE and NUE.

Our response: The reviewer has a very good point. SNUPE (soil nitrogen uptake efficiency) and NUE (nitrogen use efficiency) are different and should be described clearly in the text. As suggested by the reviewer, we added definitions for SNUPE and NUE in the Supporting Information (See track changes in the 3rd paragraph in Section "MAGPIE" in the revised SI):

"Organic N inputs to cropland soils are estimated based on the method described in (Bodirsky et al., 2014, Bodirsky et al., 2012) and include manure, crop residues, atmospheric deposition, biological

fixation by free-living microorganisms and change in soil organic matter. For pasture soils, they include manure, atmospheric deposition and biological fixation. Based on the required production, the amount of nitrogen harvested in crop biomass is estimated; subtracting the amount of N by biological fixation and N in seed provides the soil uptake. For croplands, we estimate the soil nitrogen uptake efficiency (SNUpE, Equation S5) and assume for the future an exogenous trajectory for SNUpE in line with (Kanter et al., 2020b) targets for 2030 and 2050 (see Table S3). In the policyHigh scenario, which defines only a target for 2030, we assume further improvement in SNUpE until 2050. For pastures, we estimate the nitrogen use efficiency (NUE, Equation S6). We assume that pasture NUE remains constant as in 2010 in all scenarios. Inorganic fertilizer is estimated as the necessary budget closure to fulfill plant requirements under a given SNUpE or NUE and under given quantities of organic fertilizers.

$$\begin{aligned}
 \text{SNUpE} &= \frac{\text{soil nitrogen uptake}}{\text{soil inputs}} \\
 &= \frac{\text{crops and residues} - \text{biological fixation} - \text{seed}}{\text{organic inputs} + \text{inorganic fertilizer}} \quad (\text{Eq S5}) \\
 \text{NUE} &= \frac{\text{grazed biomass}}{\text{organic N inputs} + \text{inorganic fertilizer}} \quad (\text{Eq S6})
 \end{aligned}$$

”.

Comment 16:

Supporting information: Figure S2: Spelling mistakes in figure.

Our response: We are sorry for the spelling mistakes and corrected the text in the updated figure (see updated Figure S2 in the revised SI).

Comment 17:

Supporting information: Table S1: L source says "MARINA-Nutrients Global-1.0 (developed in this study, taking inputs from MAgPIE and VIC, a detailed description is available in the next section of this file)" but Figure 1 only has VIC used in Quantity-base scarcity estimates. Please clarify.

Our response: The reviewer is right. In Figures 1 there should be an arrow from VIC to MARINA-Nutrients-Global-1.0 as well. This is because MARINA model uses hydrology from VIC to quantify nitrogen inputs to rivers and transport of N to the sub-basin outlets, as explained in the section "MARINA-Nutrients Global-1.0" of the SI. We therefore added the missing arrow in Figure 1 (see updated Figure 1 in the revised manuscript).

Comment 18:

Supporting information: Figure S5 and S6: These figures are difficult to understand. The main text is opaque from where the numbers on line 108, "24%" and "76%," come from. I suggest reworking this figure for clarity.

Our response: We regret the confusion that has arisen from the way data was presented in Figures S5 and S6. We reworked these figures and presented the data using spider charts, as shown in the updated Figures. The main purpose of these two figures, and the updated Figure 4 in the main text, is to show the differences in selected drivers/impacts (e.g., fertilizer and manure application, agricultural land, nitrogen inputs to rivers from human waste, population) of clean-water scarcity issues among the scenarios. We hope the updated figure is easier to understand.

The reviewer also has a very good comment on the basis of the numbers in line 108 of the original manuscript. We therefore added another table that includes the share of the variables (those in Figures S5 and S6) to the global total in the SI (see new Table S7 in the revised SI). We also refer to this Table in the main text (see track changes in the 3rd paragraph of Section 2.1 in the revised manuscript).

Used references in this response letter

- ALCAMO, J. & HENRICH, T. 2002. Critical regions: A model-based estimation of world water resources sensitive to global changes. *Aquatic Sciences*, 64, 352-362.
- ARNOLD, J. G., MORIASI, D. N., GASSMAN, P. W., ABBASPOUR, K. C., WHITE, M. J., SRINIVASAN, R., SANTHI, C., HARMEL, R., VAN GRIENSVEN, A. & VAN LIEW, M. W. 2012. SWAT: Model use, calibration, and validation. *Transactions of the ASABE*, 55, 1491-1508.
- ASCOTT, M. J., GOODDY, D. C., WANG, L., STUART, M. E., LEWIS, M. A., WARD, R. S. & BINLEY, A. M. 2017. Global patterns of nitrate storage in the vadose zone. *Nat Commun*, 8, 1416.
- BEUSEN, A., DOELMAN, J., VAN BEEK, L., VAN PUIJENBROEK, P., MOGOLLÓN, J., VAN GRINSVEN, H., STEHFEST, E., VAN VUUREN, D. & BOUWMAN, A. 2022. Exploring river nitrogen and phosphorus loading and export to global coastal waters in the Shared Socio-economic pathways. *Global Environmental Change*, 72, 102426.
- BODIRSKY, B., POPP, A., WEINDL, I., DIETRICH, J., ROLINSKI, S., SCHEIFFELE, L., SCHMITZ, C. & LOTZE-CAMPEN, H. 2012. Current state and future scenarios of the global agricultural nitrogen cycle. *Biogeosciences Discussions*, 9, 2755.
- BODIRSKY, B. L., POPP, A., LOTZE-CAMPEN, H., DIETRICH, J. P., ROLINSKI, S., WEINDL, I., SCHMITZ, C., MÜLLER, C., BONDSCH, M. & HUMPENÖDER, F. 2014. Reactive nitrogen requirements to feed the world in 2050 and potential to mitigate nitrogen pollution. *Nature communications*, 5, 1-7.
- CARETTA, A. M. M. A., ARFANUZZAMAN, R. B. M., MORGAN, S. M. R. & KUMAR, M. 2022. Water. In: Climate Change 2022: Impacts, Adaptation, and Vulnerability. Contribution of Working Group II to the Sixth Assessment Report of the Intergovernmental Panel on Climate Change.
- DE VRIES, W., KROS, J., KROEZE, C. & SEITZINGER, S. P. 2013. Assessing planetary and regional nitrogen boundaries related to food security and adverse environmental impacts. *Current Opinion in Environmental Sustainability*, 5, 392-402.
- DROPPERS, B., FRANSSSEN, W. H., VAN VLIET, M. T., NIJSSEN, B. & LUDWIG, F. 2019. Simulating human impacts on global water resources using VIC-5. *Geoscientific Model Development*, 13, 5029-5052.
- ELLIOTT, J., DERYNG, D., MÜLLER, C., FRIELER, K., KONZMANN, M., GERTEN, D., GLOTTER, M., FLÖRKE, M., WADA, Y. & BEST, N. 2014. Constraints and potentials of future irrigation water availability on agricultural production under climate change. *Proceedings of the National Academy of Sciences*, 111, 3239-3244.
- FINK, G., ALCAMO, J., FLÖRKE, M. & REDER, K. 2018. Phosphorus loadings to the world's largest lakes: sources and trends. *Global Biogeochemical Cycles*, 32, 617-634.
- GILBERT, M., CONCHEDDA, G., VAN BOECKEL, T. P., CINARDI, G., LINARD, C., NICOLAS, G., THANAPONGTHARM, W., D'AIETTI, L., WINT, W., NEWMAN, S. H. & ROBINSON, T. P. 2015. Income Disparities and the Global Distribution of Intensively Farmed Chicken and Pigs. *PLOS ONE*, 10, e0133381.
- GOSLING, S. N. & ARNELL, N. W. 2016. A global assessment of the impact of climate change on water scarcity. *Climatic Change*, 134, 371-385.
- GREVE, P., KAHIL, T., MOCHIZUKI, J., SCHINKO, T., SATOH, Y., BUREK, P., FISCHER, G., TRAMBEREND, S., BURTSCHER, R., LANGAN, S. & WADA, Y. 2018. Global assessment of water challenges under uncertainty in water scarcity projections. *Nature Sustainability*, 1, 486-494.
- GRIZZETTI, B., PRETATO, U., LASSALETTA, L., BILLEN, G. & GARNIER, J. 2013. The contribution of food waste to global and European nitrogen pollution. *Environmental Science & Policy*, 33, 186-195.
- HADDELAND, I., CLARK, D. B., FRANSSSEN, W., LUDWIG, F., VOß, F., ARNELL, N. W., BERTRAND, N., BEST, M., FOLWELL, S. & GERTEN, D. 2011. Multimodel estimate of the global terrestrial water balance: setup and first results. *Journal of Hydrometeorology*, 12, 869-884.
- HANASAKI, N., YOSHIKAWA, S., POKHREL, Y. & KANAE, S. 2018. A quantitative investigation of the thresholds for two conventional water scarcity indicators using a state-of-the-art global hydrological model with human activities. *Water Resources Research*, 54, 8279-8294.

- HOULTON, B. Z., ALMARAZ, M., ANEJA, V., AUSTIN, A. T., BAI, E., CASSMAN, K. G., COMPTON, J. E., DAVIDSON, E. A., ERISMAN, J. W., GALLOWAY, J. N., GU, B., YAO, G., MARTINELLI, L. A., SCOW, K., SCHLESINGER, W. H., TOMICH, T. P., WANG, C. & ZHANG, X. 2019. A World of Cobenefits: Solving the Global Nitrogen Challenge. *Earth's Future*, 7, 865-872.
- KANTER, D. R., BARTOLINI, F., KUGELBERG, S., LEIP, A., OENEMA, O. & UWIZEYE, A. 2020a. Nitrogen pollution policy beyond the farm. *Nature Food*, 1, 27-32.
- KANTER, D. R., WINIWARTER, W., BODIRSKY, B. L., BOUWMAN, L., BOYER, E., BUCKLE, S., COMPTON, J. E., DALGAARD, T., DE VRIES, W. & LECLÈRE, D. 2020b. A framework for nitrogen futures in the shared socioeconomic pathways. *Global Environmental Change*, 61, 102029.
- KOUTROULIS, A. G., PAPANIMITRIOU, L. V., GRILLAKIS, M. G., TSANIS, I. K., WARREN, R. & BETTS, R. A. 2019. Global water availability under high-end climate change: A vulnerability based assessment. *Global and Planetary Change*, 175, 52-63.
- LASSALETTA, L., BILLEN, G., GRIZZETTI, B., ANGLADE, J. & GARNIER, J. 2014. 50 year trends in nitrogen use efficiency of world cropping systems: the relationship between yield and nitrogen input to cropland. *Environmental Research Letters*, 9, 105011.
- LCWA. 2022. *Learn More About Trophic State Index (TSI) - Lake* [Online]. Available: http://www.lake.wateratlas.usf.edu/library/learn-more/learnmore.aspx?toolsection=lm_tsi [Accessed 14 June 2022].
- LI, A., WANG, M., KROEZE, C., MA, L. & STOKAL, M. 2022a. Past and future pesticide losses to Chinese waters under socioeconomic development and climate change. *Journal of Environmental Management*, 317, 115361.
- LI, Y., WANG, M., CHEN, X., CUI, S., HOFSTRA, N., KROEZE, C., MA, L., XU, W., ZHANG, Q., ZHANG, F. & STOKAL, M. 2022b. Multi-pollutant assessment of river pollution from livestock production worldwide. *Water Research*, 209, 117906.
- LIU, J., LIU, Q. & YANG, H. 2016. Assessing water scarcity by simultaneously considering environmental flow requirements, water quantity, and water quality. *Ecological indicators*, 60, 434-441.
- LIU, J., YANG, H., GOSLING, S. N., KUMMU, M., FLÖRKE, M., PFISTER, S., HANASAKI, N., WADA, Y., ZHANG, X. & ZHENG, C. 2017. Water scarcity assessments in the past, present, and future. *Earth's future*, 5, 545-559.
- LIU, K., CAO, W., ZHAO, D., LIU, S. & LIU, J. 2022. Assessment of ecological water scarcity in China. *Environmental Research Letters*, 17, 104056.
- LIU, S., XIE, Z., ZENG, Y., LIU, B., LI, R., WANG, Y., WANG, L., QIN, P., JIA, B. & XIE, J. 2019. Effects of anthropogenic nitrogen discharge on dissolved inorganic nitrogen transport in global rivers. *Global change biology*, 25, 1493-1513.
- MA, T., SUN, S., FU, G., HALL, J. W., NI, Y., HE, L., YI, J., ZHAO, N., DU, Y., PEI, T., CHENG, W., SONG, C., FANG, C. & ZHOU, C. 2020. Pollution exacerbates China's water scarcity and its regional inequality. *Nature Communications*, 11, 650.
- MEKONNEN, M. & HOEKSTRA, A. 2016. Sustainability: four billion people facing severe water scarcity. *Sci Adv* 2 (2): 1-7.
- MODI, P., HANASAKI, N., YAMAZAKI, D., BOULANGE, J. E. S. & OKI, T. 2022. Sensitivity of subregional distribution of socioeconomic conditions to the global assessment of water scarcity. *Communications Earth & Environment*, 3, 144.
- PARWEEN, S., SIDDIQUE, N. A., MAHAMMAD DIGANTA, M. T., OLBERT, A. I. & UDDIN, M. G. 2022. Assessment of urban river water quality using modified NSF water quality index model at Siliguri city, West Bengal, India. *Environmental and Sustainability Indicators*, 16, 100202.
- PASTOR, A., LUDWIG, F., BIEMANS, H., HOFF, H. & KABAT, P. 2014. Accounting for environmental flow requirements in global water assessments. *Hydrology and earth system sciences*, 18, 5041-5059.
- PRUDHOMME, C., GIUNTOLI, I., ROBINSON, E. L., CLARK, D. B., ARNELL, N. W., DANKERS, R., FEKETE, B. M., FRANSSSEN, W., GERTEN, D. & GOSLING, S. N. 2014. Hydrological droughts in the 21st century, hotspots

- and uncertainties from a global multimodel ensemble experiment. *Proceedings of the National Academy of Sciences*, 111, 3262-3267.
- SCHEWE, J., HEINKE, J., GERTEN, D., HADDELAND, I., ARNELL, N. W., CLARK, D. B., DANKERS, R., EISNER, S., FEKETE, B. M. & COLÓN-GONZÁLEZ, F. J. 2014. Multimodel assessment of water scarcity under climate change. *Proceedings of the National Academy of Sciences*, 111, 3245-3250.
- SENEVIRATNE, S. I., ZHANG, X., ADNAN, M., BADI, W., DEREZYNSKI, C., DI LUCA, A., VICENTE-SERRANO, S. M., WEHNER, M. & ZHOU, B. 2021. Weather and climate extreme events in a changing climate. In: *Climate Change 2021: The Physical Science Basis. Contribution of Working Group I to the Sixth Assessment Report of the Intergovernmental Panel on Climate Change*.
- SMIL, V. 1999. Nitrogen in crop production: An account of global flows. *Global biogeochemical cycles*, 13, 647-662.
- SPRINGMANN, M., CLARK, M., MASON-D'CROZ, D., WIEBE, K., BODIRSKY, B. L., LASSALETTA, L., DE VRIES, W., VERMEULEN, S. J., HERRERO, M. & CARLSON, K. M. 2018. Options for keeping the food system within environmental limits. *Nature*, 562, 519-525.
- STEHFEST, E., VAN VUUREN, D., BOUWMAN, L. & KRAM, T. 2014. *Integrated assessment of global environmental change with IMAGE 3.0: Model description and policy applications*, Netherlands Environmental Assessment Agency (PBL).
- STROKAL, M., BAI, Z., FRANSSSEN, W., HOFSTRA, N., KOELMANS, A. A., LUDWIG, F., MA, L., VAN PUIJENBROEK, P., SPANIER, J. E. & VERMEULEN, L. C. 2021. Urbanization: an increasing source of multiple pollutants to rivers in the 21st century. *npj Urban sustainability*, 1, 1-13.
- STROKAL, M., MA, L., BAI, Z., LUAN, S., KROEZE, C., OENEMA, O., VELTHOF, G. & ZHANG, F. 2016. Alarming nutrient pollution of Chinese rivers as a result of agricultural transitions. *Environmental Research Letters*, 11, 024014.
- STROKAL, M., SPANIER, J. E., KROEZE, C., KOELMANS, A. A., FLÖRKE, M., FRANSSSEN, W., HOFSTRA, N., LANGAN, S., TANG, T., VAN VLIET, M. T. H., WADA, Y., WANG, M., VAN WIJNEN, J. & WILLIAMS, R. 2019. Global multi-pollutant modelling of water quality: scientific challenges and future directions. *Current Opinion in Environmental Sustainability*, 36, 116-125.
- STROKAL, V., KUIPER, E. J., BAK, M. P., VRIEND, P., WANG, M., VAN WIJNEN, J. & STROKAL, M. 2022. Future microplastics in the Black Sea: River exports and reduction options for zero pollution. *Marine Pollution Bulletin*, 178, 113633.
- SUTADIAN, A. D., MUTTIL, N., YILMAZ, A. G. & PERERA, B. 2016. Development of river water quality indices—a review. *Environmental monitoring and assessment*, 188, 1-29.
- SWANEY, D. P., HOWARTH, R. W. & HONG, B. 2018. Nitrogen use efficiency and crop production: Patterns of regional variation in the United States, 1987–2012. *Science of the Total Environment*, 635, 498-511.
- TIAN, H., BIAN, Z., SHI, H., QIN, X., PAN, N., LU, C., PAN, S., TUBIELLO, F. N., CHANG, J. & CONCHEDDA, G. 2022. History of anthropogenic Nitrogen inputs (HaNi) to the terrestrial biosphere: a 5 arcmin resolution annual dataset from 1860 to 2019. *Earth System Science Data*, 14, 4551-4568.
- UDDIN, M. G., NASH, S. & OLBERT, A. I. 2021. A review of water quality index models and their use for assessing surface water quality. *Ecological Indicators*, 122, 107218.
- UDDIN, M. G., NASH, S., RAHMAN, A. & OLBERT, A. I. 2023. A sophisticated model for rating water quality. *Science of The Total Environment*, 868, 161614.
- VAN METER, K. J., BASU, N. B. & VAN CAPPELLEN, P. 2017. Two centuries of nitrogen dynamics: Legacy sources and sinks in the Mississippi and Susquehanna River Basins. *Global Biogeochemical Cycles*, 31, 2 - 23.
- VAN VLIET, M., VAN BEEK, L., EISNER, S., FLÖRKE, M., WADA, Y. & BIERKENS, M. 2016. Multi-model assessment of global hydropower and cooling water discharge potential under climate change. *Global Environmental Change*, 40, 156-170.
- VAN VLIET, M. T., FLÖRKE, M. & WADA, Y. 2017. Quality matters for water scarcity. *Nature Geoscience*, 10, 800-802.

- VAN VLIET, M. T., JONES, E. R., FLÖRKE, M., FRANSSSEN, W. H., HANASAKI, N., WADA, Y. & YEARSLEY, J. R. 2021. Global water scarcity including surface water quality and expansions of clean water technologies. *Environmental Research Letters*, 16, 024020.
- WADA, Y., WISSER, D., EISNER, S., FLÖRKE, M., GERTEN, D., HADDELAND, I., HANASAKI, N., MASAKI, Y., PORTMANN, F. T. & STACKE, T. 2013. Multimodel projections and uncertainties of irrigation water demand under climate change. *Geophysical research letters*, 40, 4626-4632.
- WANG, M., JANSSEN, A. B. G., BAZIN, J., STOKAL, M., MA, L. & KROEZE, C. 2022. Accounting for interactions between Sustainable Development Goals is essential for water pollution control in China. *Nature Communications*, 13, 730.
- WANG, M., KROEZE, C., STOKAL, M., VAN VLIET, M. T. & MA, L. 2020. Global change can make coastal eutrophication control in China more difficult. *Earth's Future*, 8, e2019EF001280.
- WANG, M., MA, L., STOKAL, M., MA, W., LIU, X. & KROEZE, C. 2018. Hotspots for Nitrogen and Phosphorus Losses from Food Production in China: A County-Scale Analysis. *Environmental Science & Technology*, 52, 5782-5791.
- YU, C., HUANG, X., CHEN, H., GODFRAY, H. C. J., WRIGHT, J. S., HALL, J. W., GONG, P., NI, S., QIAO, S. & HUANG, G. 2019. Managing nitrogen to restore water quality in China. *Nature*, 567, 516-520.
- ZENG, Z., LIU, J. & SAVENIJE, H. H. 2013. A simple approach to assess water scarcity integrating water quantity and quality. *Ecological indicators*, 34, 441-449.

REVIEWERS' COMMENTS

Reviewer #1 (Remarks to the Author):

The Authors has responded to all comments satisfactory. The current version can be accepted for the publication.

Reviewer #2 (Remarks to the Author):

The authors have effectively addressed all of my previous comments, and I am satisfied with the revisions. Therefore, I would like to recommend accepting the paper for publication.

Reviewer #4 (Remarks to the Author):

Reviewer #3 had to withdraw from the review process, so I was asked to substitute and evaluate whether the authors had addressed Reviewer #3's concerns. While I read the manuscript to fully understand Reviewer #3's comments and the authors' responses, I do not provide additional feedback beyond assessing how the authors responded to Reviewer #3's concerns.

Briefly, reviewer #3 raised concerns regarding the reproducibility of the work, and requested additional detail to allow the reader to understand how models were applied and how the MAgPIE outputs were downscaled. The authors made many changes to address Reviewer #3's concerns, and I feel they have done a good job overall. There are few aspects I feel could be elaborated on a bit further, which I outline below.

Comment 2: The authors provide several additional paragraphs to outline the MAgPIE downscaling procedure and assumptions made when applying MAgPIE. I feel the additional paragraphs mostly provide a sufficient level of detail, though there is one sentence that I feel needs to be further unpacked (lines 261-264): As the availability of organic fertilizers may be unequally distributed with availability exceeding plant requirements at places with high manure and crop residue availability, we top-up the organic by inorganic fertilizer primarily at those places where the SNU_pE (croplands) or NUE (pastures) would be highest without the addition of fertilizers.

I am struggling to understand this sentence, and it seems like it is describing a consequential assumption. More detail is needed.

Comment 3: The response to the reviewer should be added to the SI text, as it provides a clear explanation that is currently missing from the manuscript. Why are years (i.e., 2010, 2050) included for Cropland SNUPE in Table S3 but not for Pasture NUE? Note that the right-justification of the column names leads to 2050 being shown only above ssp5, when I believe it should be shown over ssp1, ssp2, and ssp5.

Comment 4: See my feedback on the response to comment 2. I feel this aspect of the work is still confusing but could be clarified with a few additional sentences.

Comment 7: While I think the authors adequately responded to this concern, I think Reviewer #3's point was that it was challenging to connect equations. Could an additional equation be included that shows how loads of DIN and DON were summed across subbasins to create L (I assume this is what was done)? This would make the connections between Squality and the MARINA equations more explicit. Or, were DIN and DON not modeled separately, and instead TDN was modeled in aggregate? Either way, adding an equation with an L term would provide clarity. If the authors do not want to add an equation, at least including some sentences to explain how L was calculated from OT and OC would be helpful.

Comment 8: A few of Reviewer #3's questions are unanswered, such as "Is all remaining N after crop uptake assumed to be exported in that year's riverine load?" I personally also found some of the details of how MARINA-Nutrients Global 1.0 was developed and applied to be lacking. For example, in Table S2, for categories 3-5, the underlying data are described as being taken directly from the MARINA-Multi-Global1.0/2.0 model or other papers, but I feel those data sources should be stated explicitly here, or perhaps included in Figure S2. The FE parameters are consequential and should be in a table.

Additionally, I think the authors addressed the point about legacy well, with one exception. The text added to the discussion reads, "Considering the historical N use can have a short-term..." Short-term should be removed from this sentence. The papers cited in this sentence point to long-term, not short-term, impacts. I would not refer to multidecadal impacts as short-term. I agree that inclusion of legacy N would increase pollution.

Response letter

Reviewer: 1

Comment 1:

The Authors has responded to all comments satisfactory. The current version can be accepted for the publication.

Our response: We thank the reviewer for recommending our manuscript for publication.

Reviewer: 2

Comment 1:

The authors have effectively addressed all of my previous comments, and I am satisfied with the revisions. Therefore, I would like to recommend accepting the paper for publication.

Our response: We thank the reviewer for recommending our manuscript for publication.

Reviewer: 4

Comment 1:

Reviewer #3 had to withdraw from the review process, so I was asked to substitute and evaluate whether the authors had addressed Reviewer #3's concerns. While I read the manuscript to fully understand Reviewer #3's comments and the authors' responses, I do not provide additional feedback beyond assessing how the authors responded to Reviewer #3's concerns.

Briefly, reviewer #3 raised concerns regarding the reproducibility of the work, and requested additional detail to allow the reader to understand how models were applied and how the MAgPIE outputs were downscaled. The authors made many changes to address Reviewer #3's concerns, and I feel they have done a good job overall. There are few aspects I feel could be elaborated on a bit further, which I outline below.

Our response: We thank the reviewer for the compliments and additional suggestions. Below we address the specific suggestions by the reviewer.

Comment 2:

About Comment 2 of Reviewer 3: The authors provide several additional paragraphs to outline the MAgPIE downscaling procedure and assumptions made when applying MAgPIE. I feel the additional paragraphs mostly provide a sufficient level of detail, though there is one sentence that I feel needs to be further unpacked (lines 261-264): As the availability of

organic fertilizers may be unequally distributed with availability exceeding plant requirements at places with high manure and crop residue availability, we top-up the organic by inorganic fertilizer primarily at those places where the SNU_pE (croplands) or NUE (pastures) would be highest without the addition of fertilizers.

I am struggling to understand this sentence, and it seems like it is describing a consequential assumption. More detail is needed.

Our response: We re-formulated the paragraph with more details about MAgPIE downscaling procedure to make this clearer (see track changes in the Section MAgPIE in the Supporting Information):

“Because no (non-modeled) global subnational data on fertilizer application exists, we assume that farmers apply fertilizer proportional to their plants’ requirements while considering the local availability of organic fertilizers. Inorganic fertilizer is distributed such that the maximum SNU_pE is minimized across the 0.5° grid cells, while allowing for a lower SNU_pE in areas where organic fertilizer is abundant. To do so, we first calculate the soil uptake requirements by the plants. We use inorganic fertilizer to top-up the organic fertilizer predominantly where the SNU_pE (croplands) or NUE (pastures) would otherwise be highest without additional inorganic fertilizers. This methodology is used to model inorganic fertilizer distribution because organic fertilizers are unequally distributed, with their availability often exceeding plant requirements in areas of high manure and crop residue availability, and their redistribution (e.g., by the transportation of manure across cells) is not simulated.”

Comment 3:

About Comment 3 of Reviewer 3: The response to the reviewer should be added to the SI text, as it provides a clear explanation that is currently missing from the manuscript. Why are years (i.e., 2010, 2050) included for Cropland SNU_pE in Table S3 but not for Pasture NUE? Note that the right-justification of the column names leads to 2050 being shown only above ssp5, when I believe it should be shown over ssp1, ssp2, and ssp5.

Our response: Our response to Comment 3 of Reviewer 3 is added in lines 196-197 of the SI: “Inorganic fertilizer is estimated as the necessary budget closure to fulfill plant requirements under a given SNU_pE or NUE and under given quantities of organic fertilizers”.

For the question: why are years (i.e., 2010, 2050) included for Cropland SNU_pE in Table S3 but not for Pasture NUE, the reason is that “We assume that pasture NUE remains constant as in 2010 in all scenarios.” This is mentioned in lines 191-192 in the SI and is now included in the table caption as well (see track changes in Table S3 in the SI).

We also modified Table S3 to make it clear that data of ssp1, ssp2, and ssp5 all refer to the year 2050 (see track changes in Table S3 in the SI).

Comment 4:

About Comment 4 of Reviewer 3: See my feedback on the response to Comment 2. I feel this aspect of the work is still confusing but could be clarified with a few additional sentences.

Our response: See our response to Comment 2 by Reviewer 3.

Comment 5:

About Comment 7 of Reviewer 3: While I think the authors adequately responded to this concern, I think Reviewer #3's point was that it was challenging to connect equations. Could an additional equation be included that shows how loads of DIN and DON were summed across subbasins to create L (I assume this is what was done)? This would make the connections between Squality and the MARINA equations more explicit. Or, were DIN and DON not modeled separately, and instead TDN was modeled in aggregate? Either way, adding an equation with an L term would provide clarity. If the authors do not want to add an equation, at least including some sentences to explain how L was calculated from OT and OC would be helpful.

Our response: We thank the reviewer for this very nice suggestion. We prefer the 2nd option suggested by the reviewer: "Including some sentences to explain how L was calculated from OT and OC would be helpful". We added the following sentences in our Method section (see track changes in lines 455-459 in the revised manuscript) "Here, L is the total dissolved N (TDN) load at the sub-basins outlets (kton/year). TDN is the sum of dissolved inorganic (DIN) and organic nitrogen (DON). DIN and DON loads at the sub-basin outlets are simulated separately by linking MAgPIE and MARINA-Nutrients (referred as $OT_{F,y,j}$ in Equation S3 for individual rivers or tributaries, and $OC_{F,y,j}$ in Equation S4 for main channel; see Figure S1 for the definition of tributary and main channel)."

Comment 6:

About Comment 8 of Reviewer 3: A few of Reviewer #3's questions are unanswered, such as "Is all remaining N after crop uptake assumed to be exported in that year's riverine load?" I personally also found some of the details of how MARINA-Nutrients Global 1.0 was developed and applied to be lacking. For example, in Table S2, for categories 3-5, the underlying data are described as being taken directly from the MARINA-Multi-Global1.0/2.0 model or other papers, but I feel those data sources should

be stated explicitly here, or perhaps included in Figure S2. The FE parameters are consequential and should be in a table.

Our response: We thank the reviewer for pointing this out. Not all remaining N after crop uptake is assumed to be exported in that year's riverine load. This is now explained in line 109 of the revised SI: "For diffuse source, $RS_{F,y,j}$ is quantified based on N input, uptake and retention of N on land as a function of runoff."

We agree with the reviewer that it is important to add sufficient details of where to find all required model inputs with their sources. For category 3 in Table S2, we added the sources of data as suggested (see track changes in the revised SI): All model inputs at the subbasin scales as shown in Figure S2, including the sources of their specific raw datasets are publicly available and can be downloaded at in the following metadata record:

<https://doi.org/10.6084/m9.figshare.13333796> (Strokhal et al., 2021). For category 4, the data source has been mentioned as van Vliet et al. (2016). For category 5, we added Wang et al. (2020) as the data source.

We agree with the reviewer the FE parameters are important. Since there are five FE parameters for each year or scenario, and for 10,336 river basins, it is thus not practical to include these numbers in a table in the manuscript. We include the FE parameters, among other required model parameters in the dataset of this paper, which will be publicly available on DANS-EASY (see the submitted readme file). We mentioned this in the revised SI in lines 138-139 "All calculated FE parameters by the model using the model inputs in Figure S2 are available in the data repository (See Data availability section in the main text)."

Comment 6:

Additionally, I think the authors addressed the point about legacy well, with one exception. The text added to the discussion reads, "Considering the historical N use can have a short-term..." Short-term should be removed from this sentence. The papers cited in this sentence point to long-term, not short-term, impacts. I would not refer to multidecadal impacts as short-term. I agree that the inclusion of legacy N would increase pollution.

Our response: We agree with the reviewer and removed "short-term" as suggested.

References:

- VAN VLIET, M., VAN BEEK, L., EISNER, S., FLÖRKE, M., WADA, Y. & BIERKENS, M. 2016. Multi-model assessment of global hydropower and cooling water discharge potential under climate change. *Global Environmental Change*, 40, 156-170.
- WANG, M., KROEZE, C., STROKAL, M., VAN VLIET, M. T. & MA, L. 2020. Global change can make coastal eutrophication control in China more difficult. *Earth's Future*, 8, e2019EF001280.